elLIFE

# KIF2A regulates the development of dentate granule cells and postnatal hippocampal wiring

Noriko Homma[1†], Ruyun Zhou[1], Muhammad Imran Naseer[2], Adeel G Chaudhary[2], Mohammed H Al-Qahtani[2], Nobutaka Hirokawa[1,2]*

[1]Department of Cell Biology and Anatomy, Graduate School of Medicine, The University of Tokyo, Tokyo, Japan; [2]Center of Excellence in Genomic Medicine Research, King Abdulaziz University, Jeddah, Saudi Arabia

**Abstract** Kinesin super family protein 2A (KIF2A), an ATP-dependent microtubule (MT) destabilizer, regulates cell migration, axon elongation, and pruning in the developing nervous system. KIF2A mutations have recently been identified in patients with malformed cortical development. However, postnatal KIF2A is continuously expressed in the hippocampus, in which new neurons are generated throughout an individual's life in established neuronal circuits. In this study, we investigated KIF2A function in the postnatal hippocampus by using tamoxifen-inducible *Kif2a* conditional knockout (*Kif2a*-cKO) mice. Despite exhibiting no significant defects in neuronal proliferation or migration, *Kif2a*-cKO mice showed signs of an epileptic hippocampus. In addition to mossy fiber sprouting, the *Kif2a*-cKO dentate granule cells (DGCs) showed *dendro-axonal conversion*, leading to the growth of many aberrant overextended dendrites that eventually developed axonal properties. These results suggested that postnatal KIF2A is a key length regulator of DGC developing neurites and is involved in the establishment of precise postnatal hippocampal wiring.

DOI: https://doi.org/10.7554/eLife.30935.001

*For correspondence:
hirokawa@m.u-tokyo.ac.jp

Present address: †Department of Lifescience, National College of Nursing, Tokyo, Japan

Competing interests: The authors declare that no competing interests exist.

## Introduction

In the mammalian nervous system, kinesin super family proteins (KIFs) play a crucial role in intracellular transport, microtubule (MT) dynamics, and signal transduction and, hence, are key players in brain function, development, and disease (*Hirokawa et al., 2010*; *Hirokawa, 1998*). KIF2A belongs to the kinesin-13 family (M-kinesin: internal motor domain family) (*Aizawa et al., 1992*; *Noda et al., 1995*; *Miki et al., 2001*), which destabilizes MTs in an ATP-dependent manner (*Desai et al., 1999*; *Howard and Hyman, 2007*; *Moores and Milligan, 2008*). In the early stages of the developing murine nervous system, KIF2A controls neurite elongation by regulating MT dynamics at neuronal growth cones and plays a crucial role in neuronal migration, axonal elongation and axon pruning in vivo (*Homma et al., 2003*; *Noda et al., 2012*; *Ogawa and Hirokawa, 2015*; *Maor-Nof et al., 2013*).

Recently, studies of KIF2A function in humans have primarily focused on cortical development because KIF2A mutations in residues Ser317 and His321 have been identified in patients with malformed cortical development (MCD) (*Cavallin et al., 2017*; *Poirier et al., 2013*). Both mutants are expected to lose MT destabilizing activity, due to a disruption in ATP binding or hydrolysis, thus resulting in a classic form of lissencephaly.

After reaching its peak in the early postnatal weeks, however, the expression of postnatal KIF2A throughout the brain is gradually restricted to specific brain regions, including the hippocampus (*Lein et al., 2007*), in which new neurons are generated throughout an individual's life in established

**eLife digest** The brain contains billions of neurons that connect together to form 'circuits' that control behavior and process information. By the time we are born, most of the neurons in our brain have already formed and connected into these circuits. But there are some brain areas that continue to make new neurons throughout our lives. One such area is the hippocampus, a region of the brain involved in learning and memory. There, neurons called dentate granule cells keep their ability to divide, migrate to new locations, and develop new connections.

Like most neurons, at the heart of each dentate granule cell is a cell body that contains the cell's nucleus and protein-making machinery. Attached to this are a set of small branch-like structures called dendrites that receive signals from surrounding neurons. Extending away from the cell body is another, longer branch called an axon, which transmits signals to other neurons.

A protein called KIF2A plays several roles in the developing brain of mammals, including helping neurons to migrate to the right place and controlling how their axons form. Before birth, neurons across the brain make KIF2A. After birth this gradually changes until only the dentate granule cells in the hippocampus produce KIF2A. In humans, mutations that prevent KIF2A from working are thought to cause brain malformations. They may also lead to disorders such as schizophrenia, epilepsy and eye defects.

To investigate the role of KIF2A in more detail, Homma et al. genetically engineered mice so that giving them a drug called tamoxifen would inactivate the gene that produces KIF2A. Mice that had this gene switched off three weeks after birth – when KIF2A levels in the hippocampus are normally at their highest – lost weight and became hyperactive. They also developed severe temporal lobe epilepsy.

To find out why these problems ocurred, Homma et al. used a microscope to study sections of the brains of the mice. The neurons had divided and migrated to the correct location of the brain with no significant problems. However, dentate granule cells that lacked KIF2A looked unusual. They had too many dendrites, the dendrites were longer than they should be and they showed markers usually only found on axons. This suggests that KIF2A helps to control the length of axons and dendrites and the wiring of the hippocampus.

At the moment, it's not known whether the same defects also occur in humans. If the results are reproducible in people, future work could help to diagnose and understand conditions linked to KIF2A, like schizophrenia and epilepsy.

DOI: https://doi.org/10.7554/eLife.30935.002

neuronal circuits (*Gonçalves et al., 2016*; *Eriksson et al., 1998*; *Kaplan and Hinds, 1977*). This postnatal expression pattern suggests that KIF2A might be involved in adult neurogenesis, neuronal migration, and the establishment of refined neuronal circuits in these brain regions, but this role has not yet been confirmed because conventional knockout mice die within 1 day of birth (*Homma et al., 2003*).

In this study, we generated tamoxifen-inducible *Kif2a* conditional knockout (*Kif2a*-cKO) mice to demonstrate the postnatal role of KIF2A in the postnatal brain. We began tamoxifen injections in postnatal week 3 (3w), when the cortical neurons or major cranial nuclei had nearly completed their migration. The 3w-*Kif2a*-cKO mice showed successful neuronal migration, but all mice died by post-natal week 6 and showed signs of hyperactivity, weight loss, and temporal lobe epilepsy (TLE). In the postnatal hippocampus, KIF2A expression was histologically restricted to the dentate mossy fibers (MFs), and the loss of KIF2A-induced MF sprouting (MFS) and aberrant recurrent excitatory circuits. In our 3w-*Kif2a*-cKO mouse model, unlike the typical TLE mouse model, the dentate granule cells (DGCs) extended aberrant axons through both the inner and outer molecular layers (IML and OML, respectively). Intriguingly, primary cultured P3-*Kif2a*-cKO DGCs did not regulate axonal or dendritic length, and consequently, the characteristics of the overextended dendrites changed, resulting in axonal conversion. These results suggested that postnatal KIF2A is a key length regulator of DGC developing neurites and is crucial for establishing postnatal hippocampal wiring.

## Results

### Weight loss, hyperactivity, and severe epilepsy were exhibited by 3w-*Kif2a*-cKO mice

To determine the role of KIF2A in the postnatal brain, we generated tamoxifen-induced *Kif2a* conditional knockout mice (*Kif2a*-cKO, *Figure 1A*). Before tamoxifen injection, these mice were normal in appearance and did not exhibit any abnormal phenotypes. We injected both wild-type (WT) and *Kif2a*-cKO siblings with tamoxifen for 7 days during the third postnatal week, after the peak expression of KIF2A in the hippocampus (*Figure 1B and C*). In addition, by the end of the second postnatal week, cortical neurons have almost finished migration and the injection timing was chosen to minimize the neuronal migratory defects in the developing cortex, which are severe in conventional knockout mice (*Homma et al., 2003*). As a result, KIF2A expression was lost in the cKO brain within 1 week of tamoxifen injection (*Figure 1D*). These cKO mice were designated 3w-*Kif2a*-cKO mice because the tamoxifen injections began at postnatal week 3. As a control for the *Kif2a*-cKO mice, WT siblings were used in all experiments after confirmation that the phenotypes of all siblings except for tamoxifen-injected *Kif2a*-cKO mice were not significantly different.

During the postnatal week 4, the 3w-*Kif2a*-cKO mice became smaller than the WT siblings (*Figure 1E*) and showed weight loss (*Figure 1F*). They also gradually developed hyperactivity (*Figure 1G*), twitching, and seizures. An open-field test showed that almost half of the 3w-*Kif2a*-cKO mice experienced an epileptic seizure within 30 min (*Figure 1H*). Eventually, all 3w-*Kif2a*-cKO mice died by postnatal day 42 (P42), the end of the postnatal week 6 (*Figure 1I*). Among them, some 3w-*Kif2a*-cKO mice died immediately after experiencing severe epileptic convulsions, which may have been one of the causes of death. The source data of body weight and activity of 3w-*Kif2a*-cKO mice and all siblings were shown in *Figure 1—source data 1*.

### The epileptic hippocampus was developed in 3w-*Kif2a*-cKO mice

To determine the focal point of the seizures, we simultaneously recorded electroencephalograms (EEGs) and behavior during postnatal week 5. The electrodes were inserted into the hippocampus and the cortex of WT and 3w-*Kif2a*-cKO siblings. In the hippocampus, 3w-*Kif2a*-cKO mice showed aberrant spikes in the EEG (arrowheads in *Figure 2A*) that coincided with twitching in the resting or locomotive state (*Figure 2B and C*). Moreover, during the epileptic seizure (*Figure 2—video 1*), the paroxysmal EEG events were clearly detected in the hippocampus but not in the cortex (*Figure 2D*). The source data of those EEG was shown in *Figure 2—source data 1*. These results suggested that the loss of postnatal KIF2A resulted in an epileptic hippocampus in 3w-*Kif2a*-cKO mice.

Supporting evidence for the epileptic hippocampus was provided by three experiments: a histological analysis of the hippocampus of 3w-*Kif2a*-cKO siblings using Bodian's silver staining method, an immunohistological analysis of frozen hippocampal sections of 3w-*Kif2a*-cKO siblings using an anti-glial antibody to detect gliosis, and a physiological analysis of the cultured hippocampal slices from P3-*Kif2a*-cKO siblings collected on P5. The third experiment required P3-Kif2a-cKO mice because P4-P6 mice should be used for hippocampal slice cultures (*Ikegaya, 1999*).

The first analysis demonstrated that, although there were no apparent laminar defects (*Figure 2E*), the DGCs of 3w-*Kif2a*-cKO mice developed many kinks and defasciculated axons in the CA3 region (*Figure 2F*), and were hypertrophic, scattered (*Figure 2G*) and swollen (*Figure 2H*) at the end of the fifth postnatal week, all possible features of hippocampal sclerosis (*Shibley and Smith, 2002*), a subsequent complication of hippocampal epilepsy. Moreover, when we began the tamoxifen injections 1 week later, at postnatal week 4, 4w-*Kif2a*-cKO mice clearly showed features of hippocampal sclerosis (*Figure 2—figure supplement 1A*) with CA1 loss (*Figure 2—figure supplement 1B*), mossy fiber sprouting (*Figure 2—figure supplement 1C*), and hypertrophic scattered DGCs (*Figure 2—figure supplement 1D*).

In the second analysis, frozen sections were stained with an anti-glial fibrillary acidic protein (GFAP) antibody because gliosis is also a sign of an epileptic hippocampus (*Pollen and Trachtenberg, 1970*; *Gupta et al., 1999*; *Loewen et al., 2016*). Importantly, the 3w-*Kif2a*-cKO hippocampus contained more GFAP-positive astroglia (*Figure 2—figure supplement 1F* and *Figure 2—figure supplement 1G*, *Figure 2—figure supplement 1—source data 1*) than the WT (*Figure 2—figure supplement 1E*).

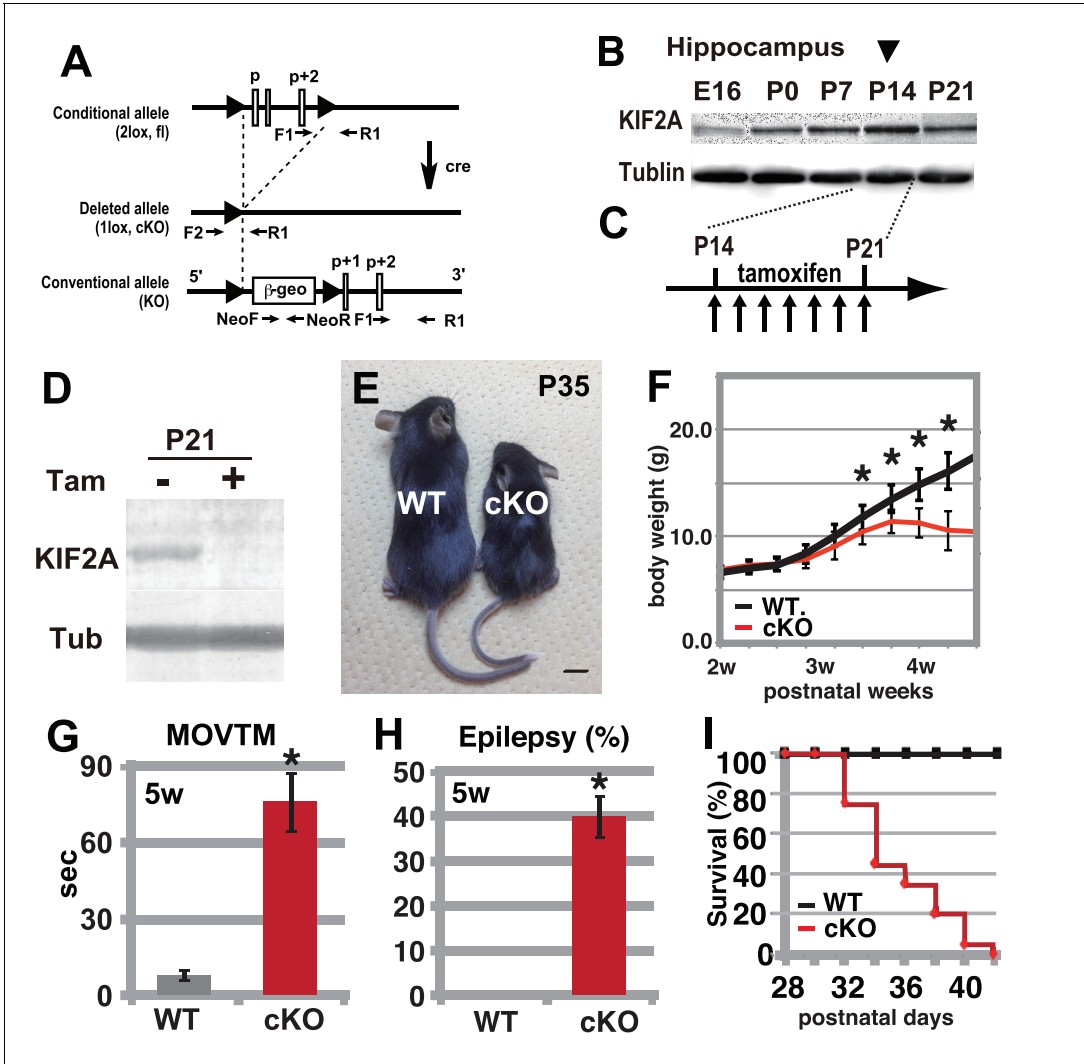

**Figure 1.** Hyperactivity, severe epilepsy, and eventual fatality in 3w-*Kif2a*-cKO mice. (**A–D**) The generation of *Kif2a*-cKO mice. (**A**) Schematic diagram of the conditional *Kif2a* targeting strategy. The conventional targeting allele is also shown at the bottom. Tamoxifen-induced Cre recombinase deletes the floxed p, p+1, and p+2 exons. (**B**). Developmental expression of KIF2A in the hippocampus. The expression peaked at P14. (**C**) The timeline for tamoxifen injection. (**D**). The loss of KIF2A. KIF2A expression was lost within 1 week after the start of the tamoxifen injection. (**E**) 3w-*Kif2a*-cKO mice at P35. The 3w-*Kif2a*-cKO mice developed smaller bodies than the WT mice. (**F**) The weight curves of WT and 3w-*Kif2a*-cKO mice. The cKO mice showed weight loss after the loss of KIF2A (n = 20, results indicate ± SD, *p<0.01; Welch's t-test). (**G**) Behavioral test. 3w-*Kif2a*-cKO mice showed hyperactivity at postnatal week 5 (n = 10, error bar indicates ± SD, *p<0.01; Welch's t-test). (**H**) The frequency of epilepsy. The 3w-*Kif2a*-cKO mice showed epileptic convulsions during a 30-min observation during postnatal week 5 (five mice each from five independent experiments, error bar indicates ± SEM, *p<0.01; Welch's t-test). (**I**) Survival rate of 3w-*Kif2a*-cKO mice (n = 20). These mice began to die starting in postnatal week 5, and all mice died by the end of week 6. Bars: 1 cm in E.

DOI: https://doi.org/10.7554/eLife.30935.003

The following source data is available for figure 1:

**Source data 1.** The raw data of body weight, activity, and survival rate of WT and 3w-*Kif2a*-cKO mice.

DOI: https://doi.org/10.7554/eLife.30935.004

In the third analysis, we attempted to demonstrate the endogenous development of excitatory recurrent circuits in the P3-Kif2a-cKO hippocampus. We dissected the hippocampus at postnatal day 5 (P5), sliced it for culturing, and performed an electrophysiological analysis at 10 days in vitro (DIV10). We then placed a stimulating electrode into the hilus and a detecting electrode into the granule cell layer (GCL) in the dentate gyrus (*Figure 2—figure supplement 1H*) to record the presence of excitatory signals, which would indicate the development of excitatory recurrent circuits in

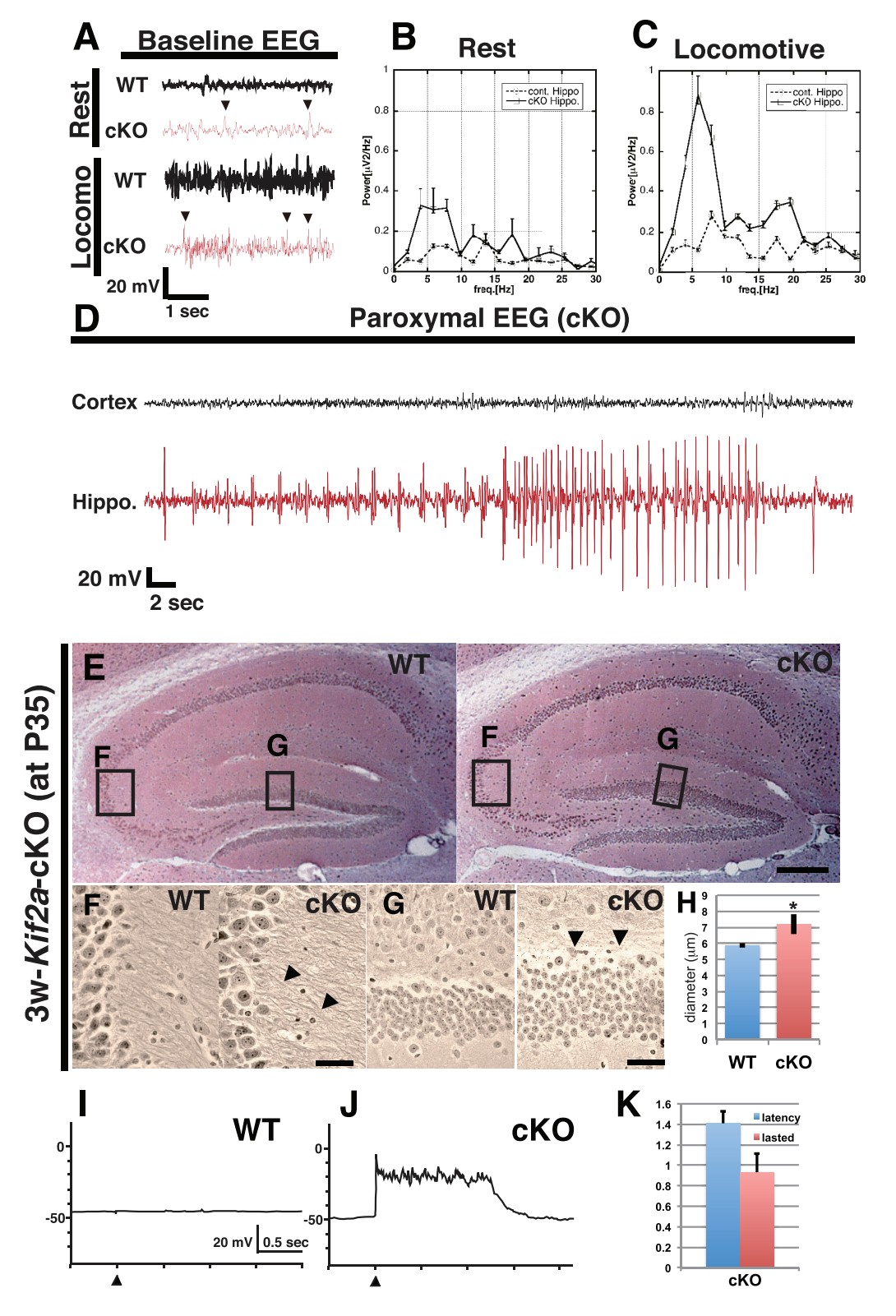

**Figure 2.** 3w-*Kif2a*-cKO mice develop epileptic hippocampi. (A–D) Electroencephalography (EEG) recordings. (A) Representative baseline EEG recordings of WT and 3w-*Kif2a*-cKO hippocampi in postnatal week 6. Some spikes were observed in the cKO brain, even in mice at rest (arrowheads). (B–C) Power spectra obtained from a fast Fourier analysis of baseline EEG recordings. Intervals of 4.5 s (for the cKO, the intervals were separated from a paroxysmal EEG recordings by at least 10 s) were selected for analysis. Three independent experiments involving two mice were performed for each

*Figure 2 continued on next page*

*Figure 2 continued*

frequency: *p<0.01 (repeated-measures ANOVA). Error bars indicate ± SEMs. (D) Representative paroxysmal EEG recordings of the cKO brain at postnatal week 4. The seizure occurred in the hippocampus. The mouse during the ictal phase of the seizure is shown in *Figure 2—video 1*. (E–H) Development of hippocampal sclerosis in a cKO brain. (E) Bodian-stained coronal hippocampus sections at postnatal week 6. (F-G) Magnified images within rectangles 'F' (CA3) and 'G' (dentate gyrus) in (E) Defasciculating fibers (arrowheads in F) and hypertrophic scattered granular cells (arrowheads in G) were observed in the cKO hippocampus. (H) Statistical analysis of the cell size. cKO DGCs are larger than the WT cells. The results are shown as the mean ± SD (7 slices each, n = 259 for WT, n = 233 for cKO). *p<0.01 (Welch's t-test). (I–K) An electrophysiological study was used to detect recurrent circuits in hippocampal primary cultures. (I–J) Representative electrographs showing data from the WT and cKO hippocampi. The black arrowhead indicates the stimulation time point. The WT slice (I) did not yield a response, and long paroxysmal depolarization shifts (PDSs) were detected in the cKO slice (J). (K) Statistical analysis of PDSs. The results are shown as the mean ± SD, n = 45, for five independent experiments involving 3 PDSs from 3 slices each. Abbreviations: ML, molecular layer; GCL, granule cell layer. Bars: 100 μm in E and 10 μm in F, G.

DOI: https://doi.org/10.7554/eLife.30935.005

The following video, source data, and figure supplements are available for figure 2:

**Source data 1.** EEG recordings of WT and 3w-*Kif2a*-cKO mice.

DOI: https://doi.org/10.7554/eLife.30935.008

**Source data 2.** The latency and lasted time of PDS occurred in the 3w-*Kif2a*-cKO hippocampal slices.

DOI: https://doi.org/10.7554/eLife.30935.009

**Figure supplement 1.** The epileptic hippocampus in tamoxifen-injected *Kif2a*-cKO mice.

DOI: https://doi.org/10.7554/eLife.30935.006

**Figure supplement 1—source data 1.** GFAP-positive area in WT and 3w-*Kif2a*-cKO dentate gyrus at P35.

DOI: https://doi.org/10.7554/eLife.30935.007

**Figure 2—Video 1.** EEG recordings of 3w-*Kif2a*-cKO mice with hippocampal epilepsy.

DOI: https://doi.org/10.7554/eLife.30935.010

the P3-Kif2a-cKO slice. As shown in *Figure 2J*, an apparent paroxysmal depolarization shift (PDS) was observed in the P3-*Kif2a*-cKO slice (*Figure 2K*, *Figure 2—source data 2*) but not in the WT hippocampus (*Figure 2I*), suggesting that the P3-Kif2a-cKO hippocampal slices had endogenously developed excitatory recurrent circuits without application of any excitatory drugs, such as picrotoxin. Together, the results suggested that recurrent excitatory circuits are endogenously induced by the loss of KIF2A without extrinsic excitation.

## Defects in cell proliferation and cell migration were not significant in the 3w-*Kif2a*-cKO hippocampus

To demonstrate how the loss of postnatal KIF2A contributes to the development of an epileptic hippocampus, we first analyzed neurogenesis and cell migration in the dentate gyrus because abnormally generated or migrated DGCs often affect epileptogenesis, seizure frequency, and seizure severity (*Korn et al., 2016*; *Hester and Danzer, 2013*; *Koyama et al., 2012*; *Houser, 1990*). Two types of thymidine analogs, 5-chloro-2'-deoxyuridine (CldU) and 5-iodo-2'-deoxyuridine (IdU), were injected for 7 days before and after tamoxifen injection to detect the newly synthesized DNA of replicating cells before and after the loss of KIF2A (*Figure 3A*). The brains of injected mice were fixed at P35, and sliced sections were stained with anti-CldU and anti-IdU antibodies. Importantly, the numbers of CldU- and IdU-positive cells were not significantly different between WT and 3w-*Kif2a*-cKO slices (*Figure 3B*, *Figure 3—source data 1*). Then, the vertical distance from the baseline of the GCL (white broken lines in *Figure 3C–3F*) to the dU-positive cells was calculated. When cells migrated into the GCL or hilus (white and blue arrowheads in *Figure 3E and F*, respectively), the distance was given a plus (+) or minus (-) value, respectively. Migration histograms show that CldU-positive cells migrated farther than IdU-positive cells, but the difference in the cellular distribution between the WT and 3w-*Kif2a*-cKO mice was not significant (*Figure 3H*, compared with 3G, see also *Figure 3I*, *Figure 3—source data 1*).

## Aberrant axon terminals of DGCs were widespread throughout the entire molecular layer in 3w-*Kif2a*-cKO mice

Before further experiments were conducted to elucidate the contribution of KIF2A to the development of an epileptic hippocampus, we analyzed the detailed distribution of KIF2A in the hippocampus by using an anti-KIF2A antibody at P21. As shown in *Figure 4A*, KIF2A expression was highly

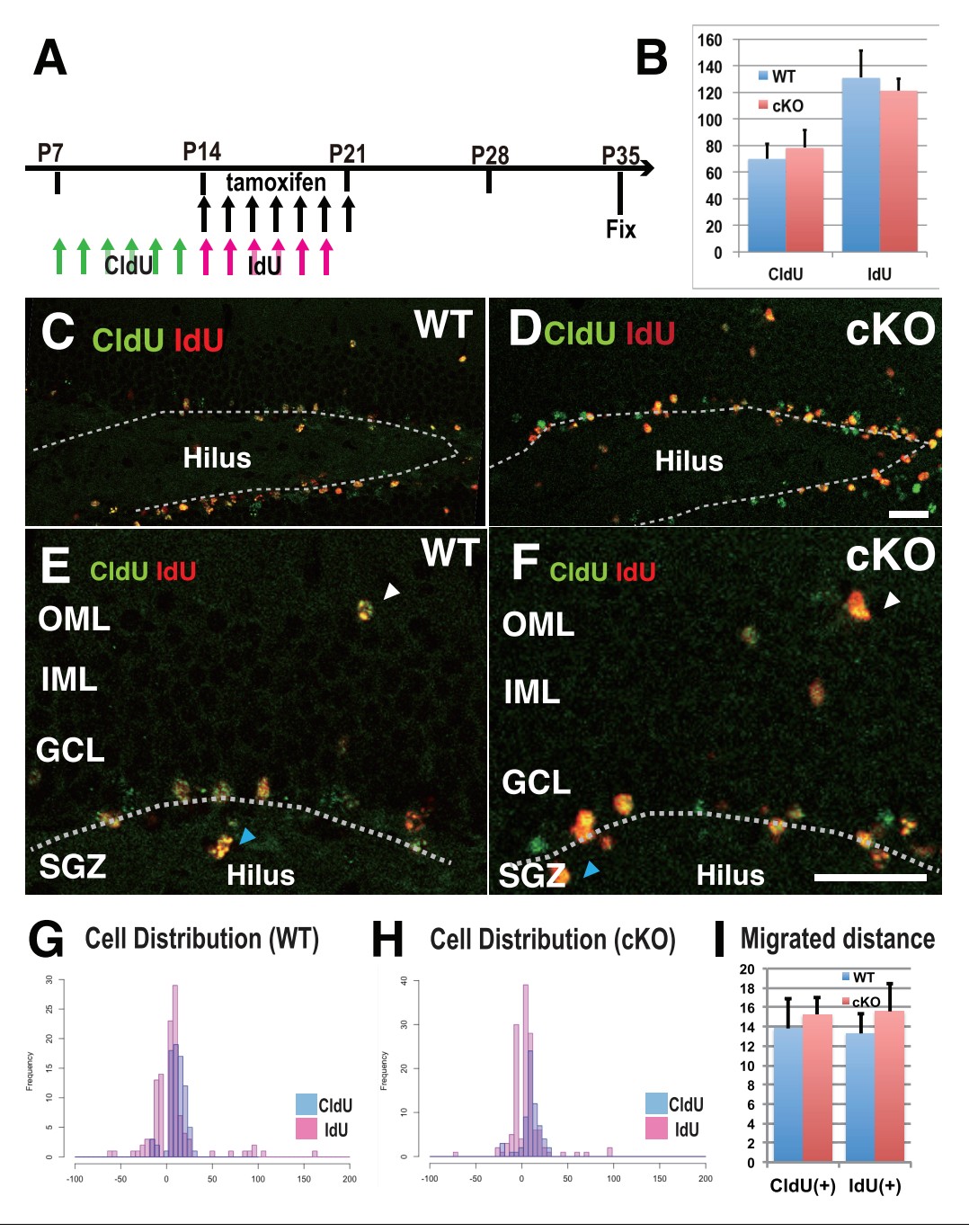

**Figure 3.** Defects in cell proliferation and cell migration were not significant in the 3w-*Kif2a*-cKO hippocampus. (**A**) The injection scheme of thymidine analogs and tamoxifen. CldU and IdU were injected during postnatal weeks 2 and 3, respectively. The mice were fixed and double-stained with anti-CldU and anti-IdU antibodies. (**B**) The number of proliferated cells before and after KIF2A loss. The results are the means ± SD. (n = 15, three independent experiments. Student' t-test, CldU p=0.61, IdU p=0.52). (**C–F**) Representative images of the immunostained dentate gyrus. E and F are magnified images of C and D. The broken white line indicates the bottom line of the GCL. Note that, in both WT (**C and E**) and cKO (**D and F**) mice, CldU-positive cells migrated slightly more than IdU-positive cells, and some aberrant cells migrated backward (blue arrowheads in the hilus) or ectopically (white arrowheads in the OML) from the bottom lines. (**G–H**) Representative graphs of the cell migration ratio. Zero represents the starting line for migrating cells at the bottom of the GCL. The distribution of CldU-positive cells and IdU-positive cells are shown in blue bars and red bars, respectively. (**I**) Statistical analysis of the cell migration ratio. The results are the means ± SEM. The average migration distance of CldU-positive cells

*Figure 3 continued on next page*

*Figure 3 continued*

(CldU(+)) and IdU-positive cells (IdU(+)) is not significantly different between WT (**G**) and cKO (**H**) cells. CldU (WT; n = 210, cKO; n = 235), and IdU (WT; n = 394, cKO; n = 364), three independent experiments. Welch' t-test, (CldU; p=0.82, IdU;p=0.89). Abbreviations: IML, inner molecular layer; OML, outer molecular layer; GCL, granule cell layer; SGZ, sub granular zone. Bars: 20 μm in D and F.

DOI: https://doi.org/10.7554/eLife.30935.011

The following source data is available for figure 3:

**Source data 1.** The migrating distance, direction, and expression of birth-dating marker of WT and 3w-*Kif2a*-cKO DGCs at P35.

DOI: https://doi.org/10.7554/eLife.30935.012

localized in the hilus and stratum lucidum of WT mice, where MFs were found (white arrowhead), whereas this effect was absent in 3w-*Kif2a*-cKO mice (*Figure 4B*). MFs are the excitatory axons of DGCs (*Watson et al., 2012*) and create synapses with their targets, which are pyramidal cells in the CA3, mossy cells in the hilus, and basket cells in the dentate gyrus (*Amaral and Dent, 1981*).

In addition, MFs are closely related to TLE as MFS is often observed in the hippocampus of patients and animal models of TLE. Thus, we hypothesized that KIF2A specifically regulates MF elongation and that the loss of KIF2A induces MFS, thus resulting in aberrant excitatory circuits and an epileptic hippocampus. Early reports have also suggested that sprouted MFs contribute to TLE pathogenesis (*Kwak et al., 2008*).

However, MFS is known to be intimately involved in the deterioration and chronicity of TLE (*Koyama et al., 2004*). When TLE occurs, excitation results in MFS, after which these collaterals recurrently elongate into the IML of the dentate gyrus where they form excitatory recurrent circuits. In short, MFS is thought to be a result of TLE.

To verify our hypothesis, we first determined the final destinations of the MFs by using Timm's staining methods. Timm sulfide silver staining is a histochemical technique used to visualize the spatial distribution of MF terminals, which specifically express high levels of $Zn^{2+}$ (*Danscher and Zimmer, 1978*). For controls, we prepared two different samples: a pilocarpine-induced TLE mouse model (*Shibley and Smith, 2002*) as a positive control for hippocampal epilepsy and a carbamazepine (CBZ)-injected 3w-*Kif2a*-cKO mouse model as a negative control in which CBZ blocks voltage-gated sodium channels and suppresses epileptic seizures during continuous use. We aimed to distinguish the effects of KIF2A deficiency from the effects of epileptic excitation on MFS.

In WT mice, Zn-positive axons of DGCs were observed in the hilus and stratum lucidum but not in the stratum oriens (so) or molecular layer (ML) (*Figure 4C*). In 3w-*Kif2a*-cKO mice (*Figure 4D*), however, the Zn-positive axons were aberrantly elongated in the stratum oriens (yellow arrowheads in *Figure 4F* compared with *Figure 4E*) and throughout the entire ML (yellow bar in *Figure 4H* compared with *Figure 4G*). Importantly, the same phenotypes were observed in the CBZ-injected 3w-*Kif2a*-cKO mice (*Figure 4—figure supplement 1B* and *Figure 4—figure supplement 1D*, compared with *Figure 4—figure supplement 1A* and *Figure 4—figure supplement 1C*, *Figure 4—figure supplement 1—source data 1*). Furthermore, the Timm grain intensities in the ML of both 3w-*Kif2a*-cKO and CBZ-injected 3w-*Kif2a*-cKO mice were >2-fold higher than those in the respective controls (*Figure 4I* and *Figure 4—figure supplement 1E*, *Figure 4—source data 1*, *Figure 4—figure supplement 1—source data 1*). Intriguingly, the signal patterns in the ML of both 3w-*Kif2a*-cKO and CBZ-injected 3w-*Kif2a*-cKO mice were different from those in the TLE mouse model in which the signal was restricted to only the IML (*Figure 4J*). These results suggested that the aberrant DGC axons of 3w-*Kif2a*-cKO mice extended throughout the ML, regardless of the presence of epileptic seizures.

Supporting those results, a different axon marker (neurofilament M, NFM), which specifically detects axons but not dendrites of DGCs in the hippocampus (*Kron et al., 2010*; *Parent et al., 1997*), and a DGC-axonal synaptic marker (synaptoporin) both exhibited wider distributions in 3w-*Kif2a*-cKO mice than in WT mice (*Figure 4—figure supplement 2B and E* compared with *Figure 4—figure supplement 2A and D*. See also *Figure 4—figure supplement 2C and F*. *Figure 4—figure supplement 2—source datas 1* and *2*).

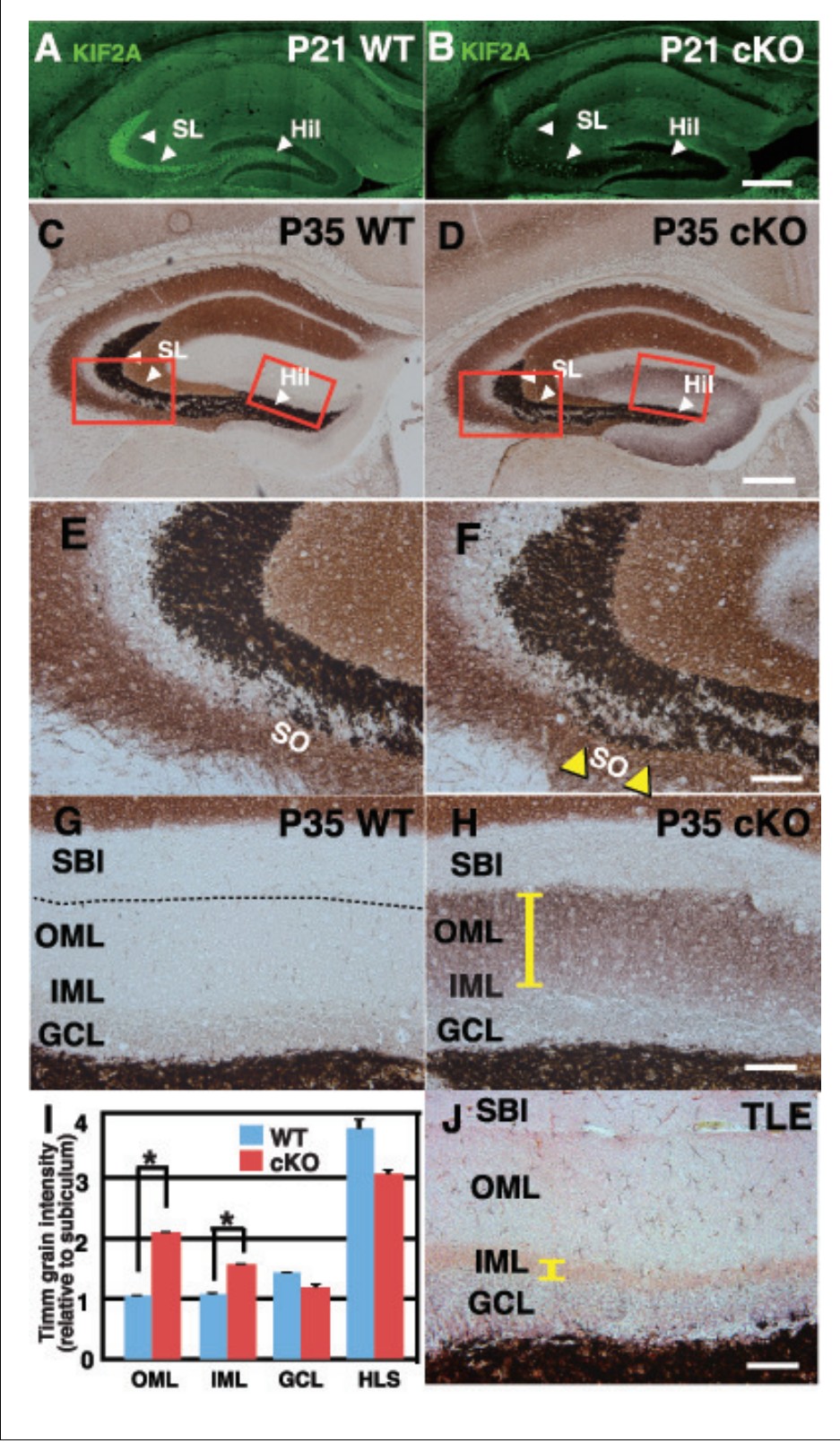

**Figure 4.** Zn-positive axon terminals of DGCs are aberrant and widespread throughout the whole ML in the 3w-*Kif2a*-cKO mice. (**A–B**) The expression of postnatal KIF2A in the hippocampus at P21. Note that KIF2A is highly expressed in the dentate hilus (Hil) and stratum lucidum (SL) (**A**) where DGCs extend their axons, termed MFs. The loss of KIF2A was observed in the cKO hippocampus (**B**). (**C–H**) Representative images showing Timm staining of

*Figure 4 continued on next page*

*Figure 4 continued*

the hippocampus at P35. (**E–H**) are magnified images indicated by the red rectangles in (**C–D**). Coronal cryosections in the hippocampus were processed for Timm histochemistry. An abnormally high dark brown density of the reaction product was observed in the SO (yellow arrowheads in **F**) and the entire dentate ML (a yellow bar in **H**) in the cKO mice compared to that in the WT mice. (**I**) Statistical analysis of Timm grain intensities. Intensities in OML and IML were significantly higher in the cKO mice than in the WT mice. The results are shown as the mean ± SEMs (5 slices each, n = 6). *p<0.01; Welch's t-test. (**J**) Positive control for the pilocarpine-induced TLE mouse model. Note that Timm reaction products are observed only in the IML (a yellow bar). Broken lines indicate the hippocampal sulcus (**G**), which forms a boundary between the ML and the SBI. Abbreviations; SL, stratum lucidum; SO, stratum oriens; SBI, subiculum; DGC, dentate granule cells; MF, mossy fiber; IML/OML, inner/outer molecular layer; GCL, granule cell layer. Bars: 400 μm in D and 100 μm in F, H, and J. See the figure supplement as well.

DOI: https://doi.org/10.7554/eLife.30935.013

The following source data and figure supplements are available for figure 4:

**Source data 1.** Timm grain intensity in WT and 3w-*Kif2a*-cKO dentate gyrus at P35.

DOI: https://doi.org/10.7554/eLife.30935.019

**Figure supplement 1.** The Zn-positive axons of DGCs abnormally extended into the SO and the entire ML in the CBZ-injected 3w-*Kif2a*-cKO mice.

DOI: https://doi.org/10.7554/eLife.30935.014

**Figure supplement 1—source data 1.** Timm grain intensity in CBZ-injected-WT and CBZ-injected-3w-*Kif2a*-cKO dentate gyrus at P35.

DOI: https://doi.org/10.7554/eLife.30935.015

**Figure supplement 2.** More DGC axons were present throughout the ML of the 3w-*Kif2a*-cKO mice than in that of WT mice.

DOI: https://doi.org/10.7554/eLife.30935.016

**Figure supplement 2—source data 1.** Intensity of NFM-staining in WT and 3w-*Kif2a*-cKO dentate gyrus at P35.

DOI: https://doi.org/10.7554/eLife.30935.017

**Figure supplement 2—source data 2.** Intensity of synaptoporin-staining in WT and 3w-*Kif2a*-cKO dentate gyrus at P35.

DOI: https://doi.org/10.7554/eLife.30935.018

## DGCs showed aberrant morphological changes in axons, cell bodies, and dendritic spines in 3w-*Kif2a*-cKO mice

To investigate the identity of the aberrant DGC axons in the entire ML, we attempted to visualize the morphology of a single DGC in the hippocampus. To this end, *Kif2a*-cKO mice were crossed with *thy1*-YFP transgenic mice (M-line) in which yellow fluorescent protein (YFP) is genetically encoded downstream of the *Thy1* promoter (*Feng et al., 2000*) and selectively expressed in a specific neuronal subset. YFP allowed for full visualization of the hippocampal neurons, including their axons, nerve terminals, dendrites, and dendritic spines. The offspring of the cross, *thy1*; YFP; *Kif2a*-cKO mice, were injected with tamoxifen beginning at postnatal week 3, and their tissues were fixed 3 weeks later. We treated 300-μm-thick sliced sections with ScaleView, an optically transparent reagent (*Hama et al., 2011*), to clarify the structure of granule cells without decreasing their fluorescence signal.

We focused on three morphological queries (*Figure 5A*). The first was whether the origin of aberrant axons in the ML of *Kif2a*-KO mice originated predominately from MFs in the hilus or directly from the cell bodies. The second was whether there were phenotypic differences between immature and mature DGCs. As shown in *Figure 5A*, after neurogenesis in the subgranular zone (SGZ), DGCs migrate into the GCL and develop a primary axon and an apical dendritic tree (*Kempermann et al., 2004*). Therefore, at postnatal week 3, immature DGCs in the inner GCL lost KIF2A in the early developmental stage, but DGCs near the ML in the outer GCL lost KIF2A after maturation. The third query related to the effects of KIF2A loss in the dendrites of mature DGCs, because alteration in MT-dynamics often affect spine morphology and function (*Jaworski et al., 2009*; *Hoogenraad and Bradke, 2009*; *Penazzi et al., 2016*).

As shown in *Figure 5B*, WT DGCs in both the inner and outer GCL (orange and red asterisks, respectively) projected a single primary axon into the hilus and extended several apical dendrites

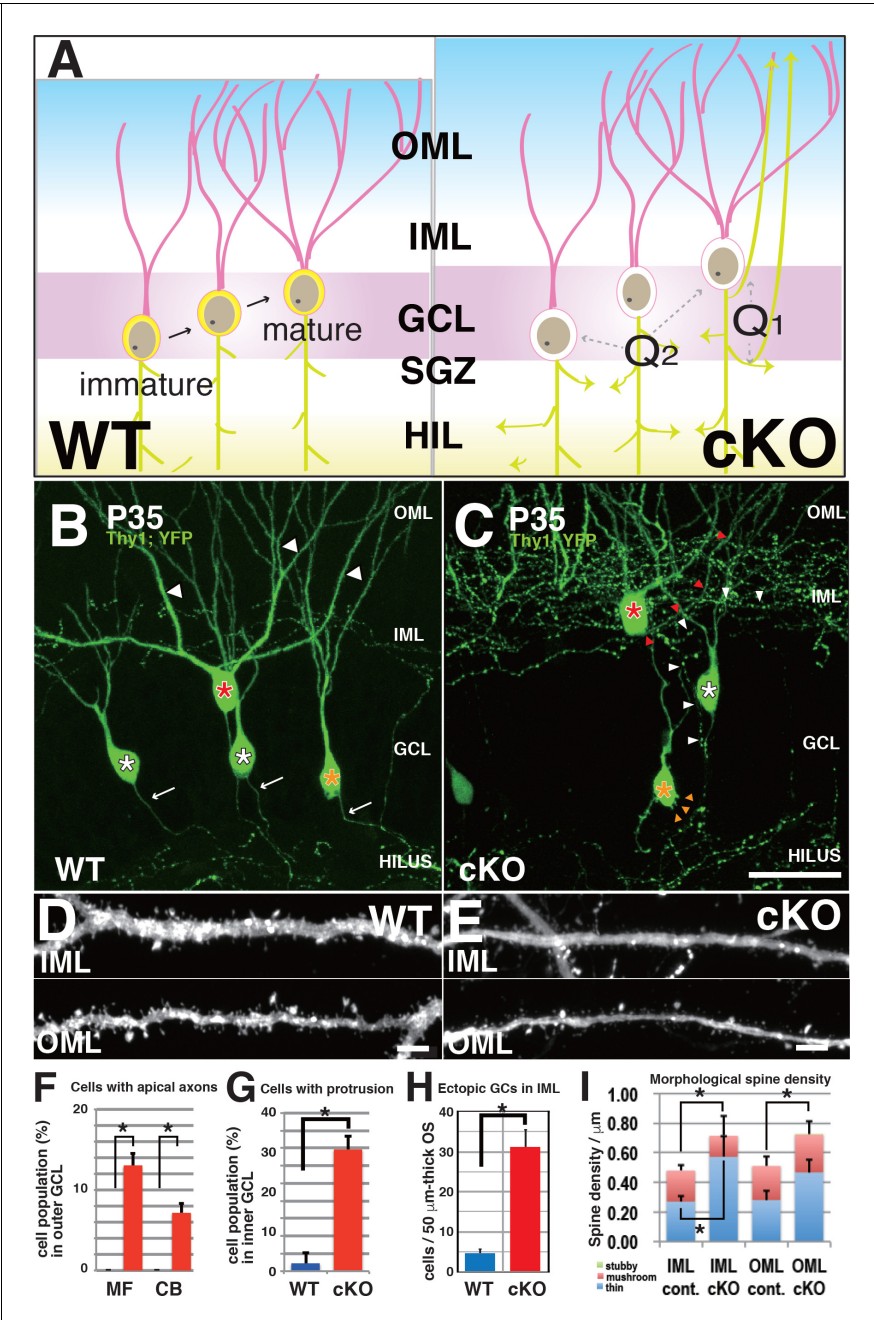

**Figure 5.** DGCs developed aberrant morphological changes in axons, cell bodies, and dendritic spines in 3w-*Kif2a*-cKO mice. (**A**) Schematics of unsolved questions in developing DGCs in the *Kif2a*-cKO dentate granule layer (DGL). In the postnatal hippocampus (left panel), DGCs continuously proliferated in the SGZ, migrated through the GCL, and then incorporated newly developed neurites into the existing hippocampal wiring. In 3w-*Kif2a*-cKO mice (right panel), the origin of overextended axons (Q1) and the difference in the immature and mature DGC phenotypes (Q2) and dendritic morphology (Q3) needs to be investigated. (**B–C**) Representative images of GFP-expressing DGCs. Z-stacked images of 300-μm-thick slices were acquired and reconstructed in 3D. In the outer area of the DGL, some mature cKO DGCs (a red asterisk in C) were aberrantly located in the IML, and developed an aberrant apical axon (red arrowheads in C), whereas WT DGCs (a red asterisk in B) extended single axons (white arrows in B) and thick dendrites (white arrowheads in B). In the middle area, some cKO DGCs (a white asterisk in C) developed recurrent axons (white arrowheads in C), whereas WT DGCs (white asterisks in B) did not. In the inner area, some cKO immature DGCs (an orange asterisk in C) developed multiple aberrant protrusions (orange arrowheads in C), whereas WT DGCs only showed a slight development of protrusions. (**D–E**) Representative images of the dendrites of matured DGCs. Z-stack images of dendrites in the IML and OML were acquired and reconstructed in 3D. Note that the spines of cKO mice appeared thinner (**E**) than those of the WT mice (**D**). (**F–H**) Statistical analysis of inner immature DGCs with apical axons (**F**) and cells with protrusions (**G**) and the number of ectopic DGCs (**H**). Cell bodies were counted in 50-μm-thick OSs (five slices each from five mice; results indicate the mean ± SDs *p<0.01, Welch's t-test). (**I**) Statistical analysis of spine density. The total spine densities were higher in both the OML and IML of cKO mice than in

*Figure 5 continued on next page*

*Figure 5 continued*
those of WT mice. Morphologically, the density of thin spines was especially increased in cKO mice compared to that in WT mice. The results indicate the mean ± SDs, n = 5, *p<0.05, Welch's t-test. Bars: 20 μm in C, and 1 μm in D and E.

DOI: https://doi.org/10.7554/eLife.30935.020

The following source data is available for figure 5:

**Source data 1.** The population of the cells with apical axons in the outer ML, the cells with protrusion in the inner ML, and the ectopic cells in inner ML in WT and 3w-*Kif2a*-cKO dentate gyrus at P35.

DOI: https://doi.org/10.7554/eLife.30935.021

**Source data 2.** The spine morphology of dendrites in IML and OML of WT and 3w-*Kif2a*-cKO dentate gyrus at P35.

DOI: https://doi.org/10.7554/eLife.30935.022

into the ML (white arrows). However, 3w-*Kif2a*-cKO mice (*Figure 5C*), which were more than 10% of outer mature DGCs, extended an aberrant axon recurrently to the ML (similarly to the cell with a white asterisk), and more than 5% of outer mature DGCs extended aberrant axons directly from the cell body (similarly to the cell with a red asterisk) (*Figure 5C and F*). In contrast, among inner immature DGCs, almost 30% of the cells had some aberrant protrusions on the cell bodies (orange arrowheads of the cell with orange asterisk in *Figure 5C and G*). Moreover, there were more ectopic DGCs in the 3w-*Kif2a*-cKO inner ML than in the ML of WT mice (*Figure 5H*). The source data of those processes were shown in *Figure 5—source data 1*.

In addition, the spine density of the dendrites of outer mature cells was higher in 3w-*Kif2a*-cKO mice (*Figure 5E*) than in WT mice (*Figure 5D*). Morphologically, the number of thin spines, not mushroom or stubby spines, was specifically higher in both the IML and OML of 3w-*Kif2a*-cKO dendrites than in those of WT dendrites (*Figure 5I*, *Figure 5—source data 2*). The results suggested that the loss of KIF2A results in more unstable or immature spines.

## Cultured *Kif2a*-cKO DGCs showed dendro-axonal conversion from DIV3

To analyze how DGCs develop and differentiate their axons and dendrites, we prepared a primary culture of dissociated DGCs from P3-*Kif2a*-cKO mice at P5, and characterized their processes with axonal markers (Tau1 or NFM) and dendritic markers (MAP2) at different stages. Before this analysis, we confirmed the DGC characteristics of the cultured cells and the loss of KIF2A from the DGCs by immunostaining the cells with anti-Prox1 (a DGC marker) and anti-KIF2A antibodies. More than 80% of cultured cells were Prox1-positive (*Figure 6—figure supplement 1A and B*), and almost all cells had lost KIF2A expression (*Figure 6—figure supplement 1C*). At DIV1, both WT and P3-*Kif2a*-cKO DGCs generated a short Tau1-dominant axon (*Figure 6B and F*) and a MAP2-dominant dendrite (*Figure 6C and G*). At DIV3, however, P3-*Kif2a*-cKO DGCs gradually generated more aberrant axons than they did dendrites. At that time, in WT DGCs, ankyrin G, the marker of the axon hillock, was detected at the neck of one axonal neurite (an arrow in *Figure 6—figure supplement 1E*). In the P3-*Kif2a*-cKO DGCs, however, ankyrin G was detected in more than one neurite (arrows in *Figure 6—figure supplement 1H*) and the population of DGCs with multiple axonal nurites was significantly larger in P3-*Kif2*-cKO DGCs than in WT DGCS (*Figure 6—figure supplement 1J*). Eventually, at DIV5, the WT DGCs developed a single long axon (a white arrowhead in *Figure 6I*) and several dendrites (a white arrow in *Figure 6K*), whereas P3-*Kif2a*-cKO DGCs developed long, defasciculated, NFM-positive axons (white arrowheads in *Figure 6J*) and generated some dendrites with multiple additional axons around their cell bodies (*Figure 6L*). A statistical analysis also demonstrated significant neogenesis of aberrant axons in P3-*Kif2a*-cKO DGCs (*Figure 6M* green bars, *Figure 6—source data 1*). These phenotypes were rescued by the KIF2A transfection (*Figure 6—figure supplement 2A–G*). The observations suggested that the neogenesis of aberrant axons in P3-*Kif2a*-cKO DGCs could be the result of a cell autonomous process rather than an altered response to the external environment, such as alterations in chemo-attraction/repulsion.

Finally, we recorded living DGCs for 24 hr at DIV2. In a WT DGC (*Figure 6—video 1*), both an axon with a single branch and a dendrite are shown to gently elongate and contract. In contrast, P3-*Kif2a*-cKO DGCs did not exhibit length control for axons or dendrites (*Figure 6—video 2*). In the video, an axon with a single branch dramatically sprouted and extended multiple branches. Even dendrites actively generated many thin spinous processes. Moreover, some protrusions instantly

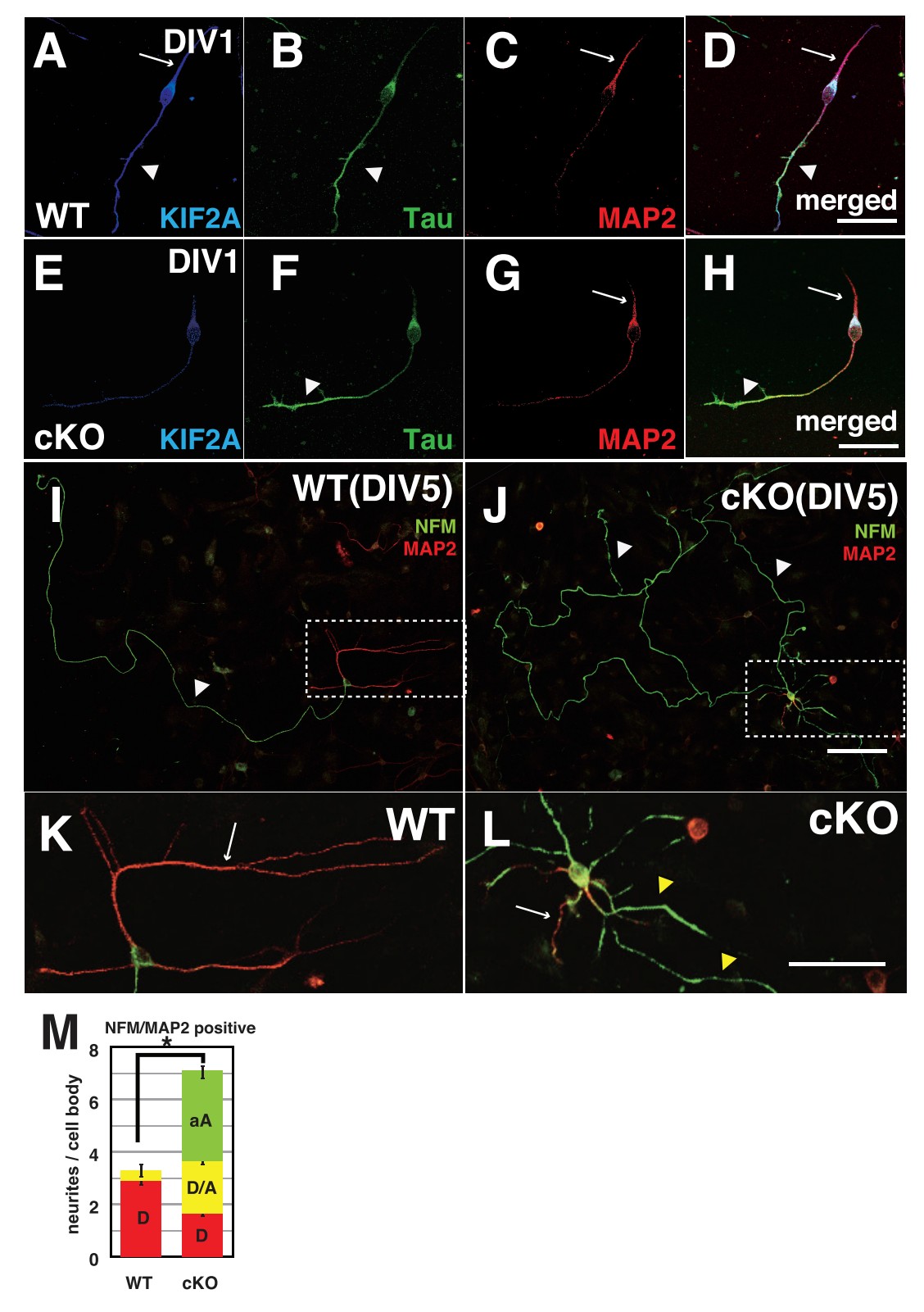

**Figure 6.** Both axons and dendrites were aberrantly developed in cultured P3-*Kif2a*-cKO DGCs. (A–L) Representative images of dissociated cultured DGCs immunostained with an axon marker (NFM or Tau1) and a dendrite marker (MAP2) at DIV1 (A–H) and DIV5 (I–L). At DIV1, both WT DGCs (A–D) and cKO DGCs (E–H) extended an axonal process (a arrowhead) and a dendritic process (an arrow). KIF2A was expressed in both processes (A compared with E). At DIV5, WT DGCs developed a single primary axon (an arrowhead in I) and several dendrites (an arrow in K), whereas cKO DGCs

*Figure 6 continued on next page*

*Figure 6 continued*

developed aberrant collateral branches from a primary axon (arrowheads in **J**) and multiple aberrant axons originating from their cell bodies (a white arrow and yellow arrowheads in **L**) in addition to some dendrites. (**K** and **L**) are magnified images of the regions indicated by the dashed squares in (**I–J**) M, DGC neuronal processes at DIV5. The numbers of processes, not including primary axons, were counted. A MAP2-positive dendrite (**D**), MAP2A- and NFM- or Tau1-positive processes (**D/A**) and NFM- or Tau1- positive aberrant axons (**A**) are shown. The results are shown as the mean ± SEMs for 20 cells each; n = 120. Error bars pointing downward indicate the SEMs for each type of neurite. The error bars pointing up indicate the SEMs for the total number of processes (*p<0.01, Welch's t-test). Bars: 100 µm in J, 50 µm in D, H, and L.

DOI: https://doi.org/10.7554/eLife.30935.023

The following video, source data, and figure supplements are available for figure 6:

**Source data 1.** The character of neuronal processes from a cell body of WT and P3-*Kif2a*-cKO DGCs.

DOI: https://doi.org/10.7554/eLife.30935.027

**Figure supplement 1.** The loss of KIF2A induced MF sprouting in dissociated cultured DGCs.

DOI: https://doi.org/10.7554/eLife.30935.024

**Figure supplement 1—source data 1.** The number of Ankyrin G-positive processes from a cell body of of WT and P3-*Kif2a*-cKO DGCs at DIV3.

DOI: https://doi.org/10.7554/eLife.30935.025

**Figure supplement 2.** Transfection of KIF2A rescued MF sprouting in dissociated cultured DGCs.

DOI: https://doi.org/10.7554/eLife.30935.026

**Figure 6—Video 1.** Time lapse recordings of cultured WT DGC.

DOI: https://doi.org/10.7554/eLife.30935.028

**Figure 6—Video 2.** Time lapse recordings of cultured P3-*Kif2a*-cKO DGC.

DOI: https://doi.org/10.7554/eLife.30935.029

appeared from the cell body, and then elongated, seemingly contacting one another. After the recording, the DGCs were fixed and stained with anti-Tau1 and anti-MAP2 antibodies (*Figure 6— figure supplement 2H and I*), revealing that the overextended dendrites had axonal (Tau1-positive, arrows in *Figure 6—figure supplement 2I*), rather than dendritic characteristics (MAP2-positive, arrowheads in *Figure 6—figure supplement 2I*). These results suggested that the loss of postnatal KIF2A might disrupt axon/dendrite determination and induce the development of multiple short axons in the hippocampus, thus resulting in complex networks in the dentate gyrus.

## Discussion

In this study, we sought to determine how KIF2A functions in the postnatal hippocampus, especially during the early postnatal weeks when DGCs are establishing a hippocampal network, because hippocampal KIF2A expression is highest in the third postnatal week (*Figure 1B*). KIF2A loss at the postnatal week 3 induced weight loss, hyperactivity, and eventually death with an epileptic hippocampus (*Figures 1E–I* and *2D–K*). In the hippocampus at P21, postnatal KIF2A was strongly expressed in dentate MF (*Figure 4A*), and its loss induced MFS (*Figure 2—figure supplement 1C*) and dysfunctional excitatory circuits (*Figure 2J*), whereas cell proliferation and cell migration were seemingly unaffected (*Figure 3*). In the 3w-*Kif2a*-cKO dentate gyrus, younger granule cells developed aberrant protrusions, and older cells developed aberrant apical axons and thinner dendritic spines (*Figure 5B–5I*). Abnormal morphogenesis of axons and dendrites was also observed in *Kif2a*-cKO cultured DGCs (*Figure 6J*). The results suggested that KIF2A plays crucial roles in the development of the precise wiring of neuronal circuits in the hippocampus by controlling the development and maintenance of the neural processes of DGCs (summarized in *Figure 7*).

### The contribution of KIF2A to the postnatal proliferation and migration of DGCs

KIF2A was predicted to play an important role in postnatal proliferation or migration, due to its effect as an MT destabilizer (*Desai et al., 1999*; *Wordeman and Mitchison, 1995*; *Manning et al., 2007*) and its critical role in proliferation (*Ems-McClung and Walczak, 2010*; *Chen et al., 2016*) and neuronal migration in the prenatal hippocampus (*Homma et al., 2003*). In this study, however, neither abnormal neurogenesis nor significant migratory defects were detected in 3w-*Kif2a*-cKO mice within 3 weeks after tamoxifen injection (*Figure 3*). Thus, from our present results, whether KIF2A is crucial for postnatal neurogenesis and migration is difficult to confirm.

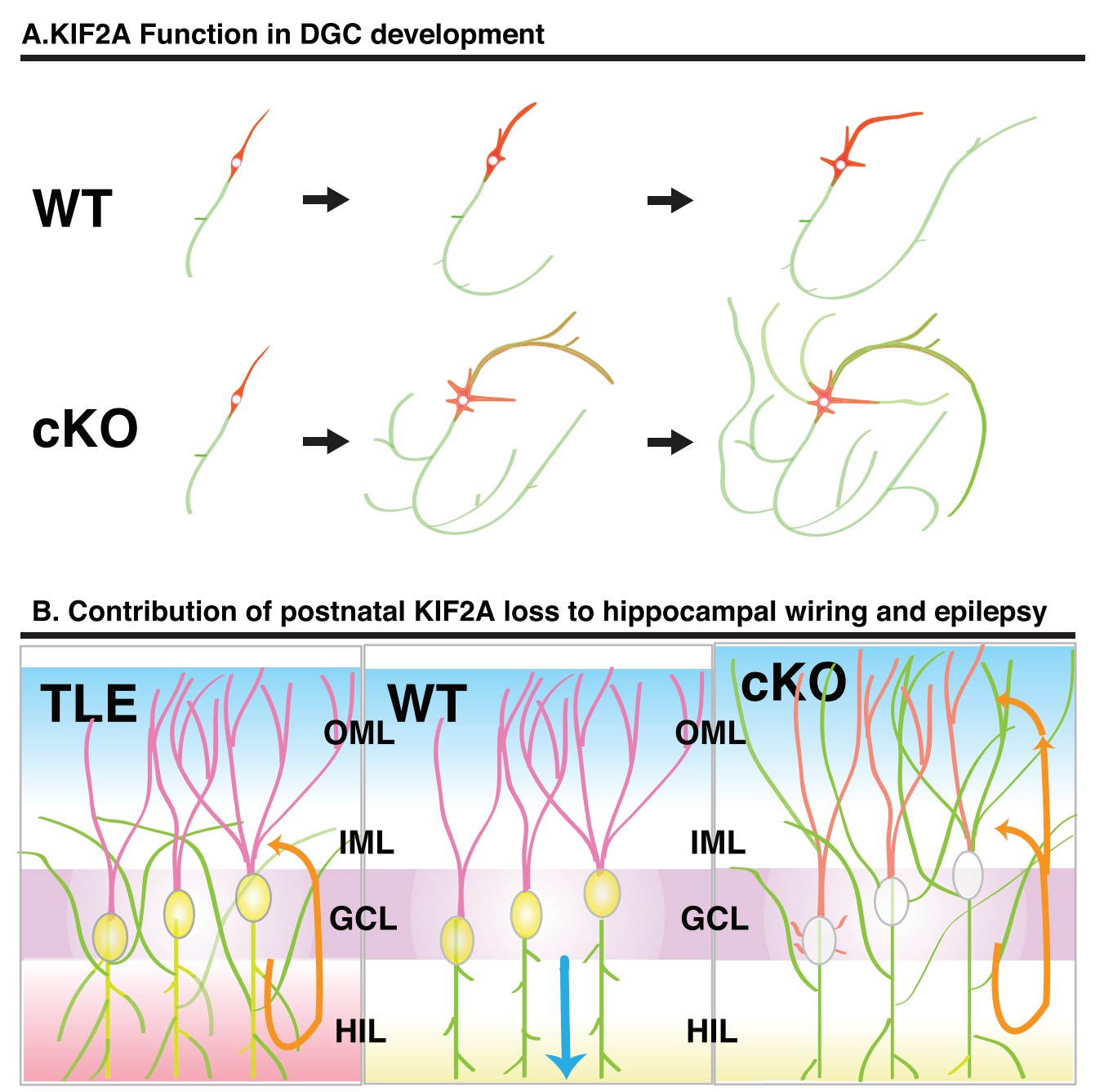

**Figure 7.** Schematic of the function of KIF2A in the postnatal hippocampus. (A) KIF2A function in DGC development. At the early stages of development, both WT and cKO DGCs normally make axonal and dendritic processes. However, whereas WT DGCs regulate the length of dendritic processes and extend a primary axon, *Kif2a*-cKO DGCs overextend and sprout both axon and dendrites, eventually developing many sprouted axonal processes. (B) KIF2A function in postnatal hippocampal wiring. In WT mice (center panel), KIF2A is highly expressed in MFs (yellow area) and actively suppresses both aberrant MFS and the elaboration of aberrant processes. The excitation of DGCs is unidirectionally transmitted along a MF (blue arrow). When the suppression is released by the loss of KIF2A (right panel), DGCs might start sprouting MFs, thus generating aberrant apical axons, and altering the dendritic features. The aberrant processes are extended into the entire ML and make reflective excitatory circuits. The excitation of DGCs is multidirectionally transmitted, and DGCs are recurrently excited (orange arrows). The repetitive excitation enhances MFS and then eventually causes TLE. In TLE mice (left panel), repetitive excitation induces MFS, but the distribution of aberrant axon terminals is limited to the IML.
DOI: https://doi.org/10.7554/eLife.30935.030

However, 3 weeks may be too short a period to allow for the detection of migratory defects in postnatal neuronal migration as the variation in the migratory distance was greater than the average DGC migration distance. The use of adult-*Kif2a*-cKO such as 8w-*Kif2a*-cKO, which can survive long enough for newborn DGCs to migrate through the entire DGC layer, may reveal the function of KIF2A in neuronal migration in the future.

## The contribution of KIF2A to the development of DGCs

Previously, KIF2A has been shown to be a key axonal collateral suppressor of prenatal hippocampal pyramidal neurons. The loss of KIF2A, a MT destabilizer, resulted in the activation of MT polymerization at the growth cone and the overextension of neuronal processes (*Homma et al., 2003*). In agreement with its prenatal functions, the expression of postnatal KIF2A was strongly distributed in the MF tract of DGCs in the hippocampus (*Figure 4A*), and the loss of KIF2A induced MFS both in vivo and in vitro (*Figures 2C*, *4F*, *6J* and *Figure 6—figure supplement 1B*), whereas KIF2A transfection rescued the aberrant collaterals (*Figure 6—figure supplement 2D*). In addition to KIF2A expression in axons, cultured DGCs expressed KIF2A in the cell bodies and dendrites at DIV1 (*Figure 6A*). The loss of KIF2A from DGCs induced the generation of multiple protrusions (*Figure 5C* and *Figure 6—video 2*) and aberrant axons from the cell bodies both in vivo and in vitro (*Figures 4H*, *5C* and *6L*), and resulted in a change in the appearance and characteristics of dendrites (*Figures 5E, I* and *6L*). These effects of KIF2A loss are interesting and may indicate a new function of KIF2A in axon/dendrite determination.

The activation of MT polymerization by collapsin response mediator protein (CRMP)−2 is known to be crucial for axon/dendrite determination and morphogenesis in hippocampal pyramidal neurons (*Bretin et al., 2005*; *Quach et al., 2015*; *Zhang et al., 2016*). Aberrant MT polymerization due to KIF2A loss might abnormally determine the fate of developing neuronal processes as axons. In other words, KIF2A may suppress the elongation of future dendrites by destabilizing MTs during the early stages of neural development.

More intriguingly, similar phenotypes of aberrant neurite growth have been associated with knockout mutations of phosphatase and tensin homologs on chromosome 10 (PTEN) in DGCs, which extended multiple long axons both in vitro and in vivo. The loss of PTEN constitutively activated Akt kinase activity (*Gary and Mattson, 2002*) and the mTORC1 pathway (*Zhou et al., 2009*). In addition, another phenotype of 3w-*Kif2a*-KO mice resembles the phenotype of *Pten*-KO mice (*Kwon et al., 2006*). *Pten*-KO mice showed signs of gliomas, microcephaly, and an epileptic hippocampus (*Luikart et al., 2011*; *Althaus and Parent, 2012*). Moreover, PTEN expression in neurons starts postnatally, and *Pten*-KO neurons developed neuronal hypertrophy and a loss of neuronal polarity. Therefore, the association of KIF2A with a PTEN-related cascade should be further elucidated. We schematically present KIF2A function in DGC development in *Figure 7A*.

## The contribution of postnatal KIF2A loss to hippocampal wiring and epilepsy

The function of KIF2A in the development of DGCs might affect postnatal hippocampal wiring because DGCs continue to proliferate in the subgranular layer (SGL), migrate through the GCL, and mature, incorporating their new processes into preliminary existing neuronal networks in the hippocampus throughout their life (*Figure 7B*, center panel) (*Gonçalves et al., 2016*; *Kaplan and Hinds, 1977*). In *Kif2a*-cKO mice (*Figure 7B*, right panel), inner immature DGCs might develop aberrant protrusions, migrating DGCs might develop aberrant axons, and outer maturing DGCs might incorporate their aberrant processes into the dentate ML and hippocampal circuits. In addition, the spine morphology is altered in dendrites lacking KIF2A. These phenotypes might mainly occur because of a lack of control of MT dynamics and might result in aberrant hippocampal wiring and epileptogenesis.

In addition, some reports have suggested that the integration of aberrantly migrated hilar ectopic granule cells into the dentate gyrus circuitry is responsible for TLE in the pilocarpine mouse model (*Cameron et al., 2011*) and in TLE patients with early-life status epileptics (*Muramatsu et al., 2008*) and febrile seizures (*Koyama et al., 2012*). In the IML in 3w-*Kif2a*-cKO mice, ectopic granule cells into the dentate morphologically integrated into the dentate ML, potentially affecting gyrus circuitry and resulting in epileptogenesis in the 3w-*Kif2a*-cKO hippocampus.

However, the influence of TLE on the *Kif2a*-cKO hippocampal phenotypes is still arguable. In ordinary TLE (left panel of *Figure 7B*), epileptic excitation induced MFS, recurrent extension of sprouted MF into the dentate IML, and aberrant excitatory recurrent circuits in the hippocampus (*McNamara, 1999*; *Gu et al., 2015*; *Liu et al., 2013*). Although we succeeded in decreasing the influence of TLE through anticonvulsant CBZ treatment in vivo (*Figure 4—figure supplement 1D*), and by reproducing the aberrant neurites in dissociated DGC cultured cells (*Figure 6*, *Figures 6—figure supplement 1* and *2*), this approach must still be carefully evaluated when examining the contribution of the loss of KIF2A to the epileptogenesis and phenotypes of 3w-*Kif2a*-cKO mice. For example, regarding the expansion of the aberrant axons throughout the ML in 3w-*Kif2a*-cKO mice, in TLE, many other types of axons in the hippocampus in addition to MFs exhibit sprouting, but those axons are not detected by Timm's stain. In 3w-*Kif2a*-KO mice, MFS may extend into the OML simply because other types of axons did not sprout in our model, thus leaving more 'space' for MF sprouting. However, we believe that the 'space' might not be greatly different between 3w-*Kif2a*-cKO mice (*Figure 4H*) and the 3w-induced TLE animal models (*Figure 4J*) because both mice showed TLE beginning at postnatal week 4, and the sprouting of many types of axons, including MFs in the hippocampus, might be induced by TLE in the ML at the same level.

Regarding the regulation of KIF2A function, the phenotypes of tamoxifen-injected *Kif2a*-KO DGCs suggested that KIF2A may possibly contribute to MFS in ordinary TLE. After epileptic fits in the hippocampus, hyper-excitation locally up-regulates brain-derived neurotrophic factor (BDNF) in the target site of the MF projection, the stratum lucidum (SL). The local activity of BDNF in the hilus initiates MFS, thus eventually resulting in hippocampal hyper-excitability (*Koyama et al., 2004*; *Tamura et al., 2009*). As shown in *Figure 4A*, KIF2A is highly expressed in the SL, and previous research has shown that BDNF-derived kinases p21-activated kinase 1 (PAK1) and cyclin-dependent kinase 5 (CDK5) block the function of KIF2A by phosphorylation in cortical neurons (*Ogawa et al., 2012*); thus, the hyper-excitation-induced local activation of BDNF might potentially block KIF2A function and therefore induce MFS only in the hilus. Detailed analysis of tamoxifen-injected *Kif2a*-KO DGCs would demonstrate the contribution of KIF2A to MFS in ordinary TLE.

## The contribution of KIF2A loss to weight loss and hyperactivity

The 3w-*Kif2a*-cKO mice exhibited weight loss (*Figure 1F*) and hyperactivity (*Figure 1G*), first becoming hyperactive, then gradually losing interest in food, becoming weak, and eventually dying. The link between these phenotypes and epilepsy remains unclear, but some reports have suggested that these phenotypes are associated with hippocampal dysfunction (*Fallet-Bianco et al., 2008*). Doublecortin (DCX)-KO mice, which exhibit both weight loss and hyperactivity, harbor a neuronal lamination defect in the hippocampus (*Håvik et al., 2012*). Tuba1a-KO mice, which are hyperactive, exhibit prominent hippocampal lamination defects (*Fallet-Bianco et al., 2008*). Although 3w-*Kif2a*-cKO mice did not show an apparent hippocampal laminar defect at P35 (*Figure 2E*), ectopic cells in the dentate gyrus or pyramidal cells displaced by sprouted MFs would result in weight loss and hyperactivity.

Moreover, both phenotypes have also often been reported in MCD patients and animal models (*Stottmann et al., 2013*; *Stottmann et al., 2017*). Four recently reported *KIF2A*-mutated human pediatric patients with MCD display band heterotopia, posterior predominant pachygyria, a thin corpus callosum, severe congenital microcephaly, and neonatal-onset seizures (*Poirier et al., 2013*). Because the patients were young at the time of the report (3 and 5 months old), further research is necessary to confirm the hippocampal phenotypes of *Kif2a*-mutated MCD patients.

## Future perspectives

We herein demonstrated that postnatal KIF2A regulates the development of DGCs and the wiring of neuronal circuits in the hippocampus. However, the relevance of the hippocampal phenotypes of *Kif2a*-KO mice to human patients with mutations in *Kif2a* is not certain. In the future, the molecular mechanisms of KIF2A regulation in DGC development and hippocampal wiring should be explored in both KO mice and in human patients. The progress of this line of research will allow for analysis of the pathogenesis of *Kif2a*-related diseases, including schizophrenia (*Li et al., 2006*), juvenile myoclonic epilepsy (*Kapoor et al., 2007*), mental retardation, ocular defects (*Jaillard et al., 2011*), and

MCD (*Poirier et al., 2013*). We hope that the collection of data on KIF2A-deficient mice will clarify the pathogenesis of these diseases and lead to a more accurate diagnosis in humans.

# Materials and methods

## Key resources table

| Reagent type (species) or resource | Designation | Source or reference | Identifiers | Additional information |
|---|---|---|---|---|
| Gene (*Mus musculus*) | Kif2a | PMID:1447303 | MGI:108390 | |
| Strain, strain background (*Mus musculus*) | C57BU6NJcl, C57BU6JJcl | Clea Japan | | |
| Strain, strain background (*Mus musculus*) | CreERt(CAG-cre/Esr1) | The Jackson Laboratory | Tg(CAG-cre/Esr1 *)5Amc/J (Stock No: 004453) | |
| Strain, strain background (*Mus musculus*) | R26R (lacZ expression with the ROSA26 Cre reporter strain) | PMID:9916792 | | |
| Strain, strain background (*Mus musculus*) | Thy1-EGFP line M | The Jackson Laboratory | Tg(Thy1-EGFP)MJrs/J (Stock No: 007788) | |
| Cell line (Mus musculus) | CMTl-1 | Merck Millipore | 129/Sv-derived ES cell (ESC) line | |
| Antibody | anti-KIF2A (mouse monoclonal) | PMID: 7535303 | | (1:200) |
| Antibody | anti-KIF2A (rabbit polyclonal) | Thermo Fisher | Thermo Fisher: PA3-16833 | (1:1500) |
| Antibody | anti-GFAP(mouse monoclonal) | BO Transduction Laboratory | BO Bioscience: 610565 (Clone 52/GFAP) | (1:200) |
| Antibody | anti-BrdU (mouse monoclonal) for IdU | BO Pharmingen | BO Bioscience: 555627 (Clone 304) | (1:1000) |
| Antibody | anti-BrdU (rat monoclonal) for CldU | Accurate Chemicals | Accurate Chemicals (Clone: BU1/75) | (1:500) |
| Antibody | anti-Tau1 (mouse monoclonal) | Merck Millipore | Merck Millipore: MAB3410 (Clone PC1C6) | (1:1000) |
| Antibody | anti-Neurofilament M (mouse monoclonal) | Sigma | Sigma: N5264 (Clone NN18) | (1:600) |
| Antibody | anti-MAP2 (chicken monoclonal) | Abeam | Abeam: AB5392 (Clone 304) | (1:200) |
| Antibody | anti-Synaptoporin (rabbit polyclonal) | Synapitic Systems | SYSY:102002 | (1:300) |
| Antibody | anti-Prox1 (rabbit polyclonal) | Merck Millipore | Merck Millipore: AB5475 | (1:10000) |
| Antibody | anti-Ankyrin G (mouse monoclonal) | Thermo Fisher | Thermo Fisher: 338800 (Clone 4G318) | (1:600) |
| Antibody | Alexa 488-, 546-, 555-, 647, (Goat polyclonal) | Molecular Probes | | (1: 200–1: 500) |
| Recombinant DNA reagent | pEYFP-C1 (vector) | Clontech | | |
| Recombinant DNA reagent | YFP-Kif2a (full length) | PMID: 14980225 | | Progenitors: pEYFP-C1 vector |
| Commercial assay or kit | LA PCR Kit | Takara | TaKaRa:RR013A | |
| Chemical compound, drug | Tamoxifen | Sigma | Sigma:T5648 | (30 mg/kg body weight/day) |
| Chemical compound, drug | Pilocarpine | Sigma | Sigma: P6503 | (290 mg/kg body weight/day}, |
| Chemical compound, drug | Scopolamine methyl bromide | Sigma | sigma: 8502 | (1 mg/kg body weight/day) |
| Chemical compound, drug | ScaleView | Olympus | | |
| Chemical compound, drug | 5-iodo-2'-deoxyuridine | Sigma | Sigma: 17125 | 50 mg/kg |
| Chemical compound, drug | 5-chloro-2 '-deoxyuridene | Sigma | Sigma: C6891 | 50 mg/kg |
| Chemical compound, drug | poly-L-lysine | Sigma | Sigma: P-2636 | 100 mg/ml |
| Chemical compound, drug | Laminin | Invitrogen | Invitrogen: 23017–015 | 25 µg/ml |
| Software, algorithm | MicroMax 1. 3 | AccuScan Instrument | | |

*Continued on next page*

*Continued*

| Reagent type (species) or resource | Designation | Source or reference | Identifiers | Additional information |
|---|---|---|---|---|
| Software, algorithm | Clampex 9.2 software | Axon Instruments | | |
| Software, algorithm | IMARIS 7.2 | Bitplane:Zeiss | | |
| Software, algorithm | ImageJ 1.51 hr | NIH | | |
| Software, algorithm | SPSS V22 | IBM | | |
| Machine | MicroMax Monitor | AccuScan Instruments | | |
| Machine | Axopatch 1D amplifier | Axon Instruments | | |

## Conditional gene targeting of the *Kif2a* gene

A 3loxP-type targeting vector was constructed by using a genomic clone obtained from an EMBL3 genomic library, and genomic fragments were amplified from the 129/Sv-derived ES cell (ESC) line CMT1-1 (Chemicon/Millipore, Billerica, MA) by using an LA-PCR kit (Takara, Japan). The CMT1-1 ESCs were transfected with the targeting vector and screened for homologous recombinants using PCR. The 3loxP/+ESCs were electroporated using a pCre-Pac plasmid to remove the selection cassette flanked by loxP sequences. The 2loxP/+ESCs were injected into blastocysts, and chimeric male mice were obtained and bred with C57BL/6J female mice. Germline transmission was confirmed by PCR using tail DNA samples. $Kif2a^{fl/fl}$ mice were produced by an intercross with $Kif2a^{fl/+}$ mice. To conditionally delete exons flanked by loxP and driven by the chicken beta-actin promoter/enhancer coupled with the cytomegalovirus (CMV) enhancer (CBA), tamoxifen-inducible Cre transgenic mice (CreERt; CAG-Cre/Esr1; Jax #004453, JAX MICE Laboratories, Bar Harbor, ME) were used. The CBA-CreERt mice were characterized by using lacZ expression with ROSA26 reporter mice (R26R), which have a loxP-flanked STOP sequence followed by the lacZ gene inserted into the gene trap ROSA26 locus (By courtesy of Prof. Soriano [*Soriano, 1999*]). Conventional knockout mice (*Homma et al., 2003*) were crossed with CBA-CreERt$^{+/-}$ mice to obtain $Kif2a^{+/-}$;CBA-CreERt$^{+/-}$ mice. Male $Kif2a^{+/-}$; CBA-CreERt$^{+/-}$ mice were mated with female $Kif2a^{fl/fl}$ mice to produce offspring that contained the $Kif2a^{fl/+}$, $Kif2a^{fl/+}$; CBA-CreERt$^{+/-}$, $Kif2a^{fl/-}$, and $Kif2a^{fl/-}$; CBA-CreERt$^{+/-}$ alleles. Tamoxifen (Sigma, St. Louis, MO, 10 mg/ml) was dissolved in sunflower oil and administered at a dose of 30 mg per kg body weight daily for 7 consecutive days. The *Kif2a* deletion occurred when the tamoxifen-induced Cre recombinase deleted the floxed DNA domain, which was followed by a frameshift during the *Kif2a* RNA translation. Deletion was confirmed by a western blot analysis of the crude extracts of whole brain tissues at P21 by using a monoclonal antibody against the N-terminal region of KIF2A (*Noda et al., 2012*). For control mice, we generally used wild-type mice after ensuring that the $Kif2a^{fl/-}$; CBA-CreERt$^{+/-}$ mice and WT mice were not significantly different. The genotypes were determined by PCR of tail DNA or DNA from ES cells with the following primers (see *Figure 1A*):

F1, 5'-CGCTCATGTGTTTTAAGCTG-3';

R1, 5'- CACCCCACTATAACCCAGCATTCG-3';

F2, 5'-GCTGCCAGTGACATAGACTAC-3', and the Neo and Cre transgenes. The mice were maintained by repeated backcrossing with C57BL/6J mice (>12 times) in a pathogen-free environment.

## TLE model mice

The mice received an intraperitoneal (i.p.) injection of scopolamine methyl bromide (Sigma, St. Louis, MO, 1 mg/kg) in a sterile saline vehicle (0.9% NaCl, 0.1 ml total volume) 30 min prior to an injection of pilocarpine to decrease the peripheral cholinergic effects of pilocarpine. The experimental animals were then i.p. injected with a single dose of pilocarpine (Sigma, 290 mg/kg), as previously described (*Shibley and Smith, 2002*). The WT mice were age-matched with treated mice and received a comparable volume of vehicle.

## Behavior tests

WT male mice and 3w-*Kif2a*-cKO (P25 littermates) were used in all behavioral tests in a blinded manner. The home cage activity tests were conducted using a MicroMax Monitor (AccuScan Instruments, Columbus, OH) and quantified using a computer-operated MicroMax 1.3 (AccuScan Instrument).

The monitor displayed 16 invisible infrared light beams per axis with synchronous filtering, double modulation and digital hysteresis. These beams provide information that describes the movement of an animal in its home cage, thus allowing an animal's behavior to be monitored. Mice that were housed singly in their home cages were placed in the beam boxes for 5 min, and their activity was continually recorded. The measurements used to assess home cage activity included active time. The average amount of active time was analyzed using Student's t-tests. For epilepsy, five mice were isolated in a cage and observed for 30 min. The epileptic mice were genotyped after the observation.

### EEG recording

WT and 3w-*Kif2a*-cKO siblings were anesthetized in the postnatal week 4 by using ketamine/xylazine and were surgically implanted with a set of electrodes. Two 0.1 mm diameter silver wires were bonded, including a 1.2-mm-long reference electrode and a 2.0-mm-long working electrode with a hard epoxy resin coat (except for its 0.2-mm-long exposed tip), which served to electrically insulate the probe from the reference electrode. Dental cement (GC Dental, Tokyo, Japan) was used to fix the electrode set to the skull. The electrode positions in the left hemisphere and the CA1 of the left hippocampus were stereotaxically determined as 1.3/1.3 mm or 2.0/1.8 mm anterior to the bregma and 1.2/1.2 mm or 1.5/1.5 mm lateral to the midline at a depth of 1.3/1.2 mm or 1.5/1.3 mm for the WT or 3w-*Kif2a*-cKO mice, respectively. These differences were due to the differences in the average brain sizes between the two genotypes. EEG recordings were obtained from mice after complete recovery. The electrodes, measurement system, and software were all purchased from Unique Medical (Tokyo, Japan). EEG recordings were obtained from five mice for each genotype. After EEG recordings, we confirmed the electrode position using a histological examination.

### Electrophysiology

The patch-clamp recordings of DGCs were obtained at room temperature using an Axopatch 1D amplifier (Axon Instruments, Union City, CA). Patch pipettes (3–5 M$\Omega$) were filled with 122.5 mM Cs gluconate, 17.5 mM CsCl, 10 mM HEPES, 0.2 mM EGTA, 8 mM NaCl, 2 mM Mg-ATP, and 0.3 Na-GTP (pH 7.2, 290–300 mM mOsm). A slice was transferred to a recording chamber and continuously perfused with cold oxygenated ACSF containing 119 mM NaCl, 2.6 mM KCl, 1.3 mM MgSO$_4$, 1 mM NaH$_2$PO$_4$, 26 mM NaHCO$_3$, 2.5 mM CaCl$_2$, and 11 mM D-glucose. Single pulse stimuli were delivered by bipolar tungsten electrodes, which were positioned on the hilus far from the recorded cells, to avoid antidromic activation. Absence of antidromic activation contamination was concluded if CNQX-AP-5 completely eliminated any responses to the stimulus. The signals were filtered at 2 kHz, digitized at 10 kHz, and analyzed using Clampex 9.2 software (Axon Instruments, Union City, CA).

### Histological analysis

The nervous elements were stained using the standard Bodian method (*Bodian, 1936*; *Bodian, 1937*). Briefly, the brains were fixed in FEA (formalin-ethanol-acetic acid: 90 ml of 80% ethanol with 5 ml of formaldehyde and 5 ml of glacial acetic acid), dehydrated with ethanol, and embedded in paraffin. The tissue was sectioned at a thickness of 7 μm. The paraffin sections were hydrated in distilled water (DW), and the slides were then incubated in 2% Protargol solution for 48 hr at 37°C in the dark with 5 g of polished copper shot. The samples were then rinsed three times with DW; reduced in 1% hydroquinone for 10 min; rinsed three times with DW; immersed in 1% aqueous gold chloride for 10 min; rinsed three times with DW; developed in 2% oxalic acid for 20 min; rinsed twice with DW; fixed in 5% sodium thiosulfate for 5 min; rinsed five times with DW; dehydrated; and mounted with cover slips.

For brain tissue immunohistochemistry, the mice were perfused with a solution of 4% paraformaldehyde (PFA) and 0.1% glutaraldehyde (GA) in 0.1 M sodium phosphate buffer (PB, pH 7.4). Subsequently, 30-μm-thick frozen slices were rinsed in PBS, fixed for 10 min, permeabilized with 0.1% Triton X-100 in PBS for 10 min, incubated in blocking solution (5% normal goat serum in PBS) for 30 min, and incubated with primary antibodies at 4°C overnight. After the tissues were washed with PBS, secondary antibodies were applied at 4°C overnight. To stain thick sections, 0.1% Triton X-100 was added to the blocking solution. All antibodies were as described in the previous section. To visualize YFP-expressing GCs in mouse brains, the mice were perfused with a fixation solution, and

300-μm-thick sections were immersed in ScaleView (Olympus, Japan), an optically transparent reagent, at 4°C for 24 hr. For all experiments, we used littermates for controls and selected slices at comparable positions determined with a brain atlas. The samples were observed under an LSM710 or LSM 780 confocal microscope (Zeiss).

## Birth-dating analysis

Immunofluorescence detection of two thymidine analogues (CldU and IdU) was performed along with Tuttle's methods (*Tuttle et al., 2010*). Briefly, 5-iodo-2'-deoxyuridine and 5-chloro-2'-deoxyuridene (IdU and CldU; Sigma, St. Louis, MO) were dissolved in saline at 10 mg/ml as stock solution. Proliferating cells in the hippocampus were labeled by sequential intraperitoneal injection at 50 mg/kg for 1 week before or after tamoxifen injection. All mice were perfused with 4% PFA/PBS, dehydrated with ethanol, and embedded in paraffin. The 7-μm-thick sections were rehydrated with ethanol, washed for 5 min in PBS, and permeabilized with 0.1% Triton-X 100 for 5 min; then, antigen retrieval was performed in boiling 0.01 M pH 6.0 sodium citrate buffer for 20 min by using a microwave oven. Slides were immersed in 1.5 N HCL for 40 min at RT, washed twice in PBS for 5 min, and immersed in blocking solution (5% goat serum in PBS). Then an anti-IdU antibody diluted in blocking solution was applied and incubated overnight at 4°C. After being agitated in PBS for 20 min in a shaking jar at 37°C, the slices were washed four times in PBS, and anti-CldU antibodies diluted in blocking solution were applied and incubated overnight at 4°C. The slides were washed twice for 5 min per wash in PBS, and then, the appropriate secondary antibody solution (1:300) was applied for 2 hr at RT. The slices were washed 5 times for 5 min per wash in PBS, and a cover glass was applied with PBS.

The distance between the bottom edge of the GCL and the dentate granule cells was measured using IMARIS software (Zeiss).

## Timm staining

To visualize zinc and other metals in the hippocampus, 30-μm-thick frozen brain sections were stained using the neo-Timm method with some modifications (*Danscher and Zimmer, 1978*). The pixel intensities were measured as previously described (*Koyama et al., 2004*). Briefly, in an image acquired using a 20 × objective, at least five 20 μm × 20 μm cursor points at 20 μm intervals were positioned in each hilus, granular and IML, OML, and subicular area located just outside the hippocampal sulcus. The mean signal intensity (I) within these cursor points was measured at an 8-bit resolution using ImageJ software (NIH, USA). The Timm grain intensity was determined by dividing the values of these subregions by the value of the subiculum (background). The same method was used to measure the NFM intensity. As an internal control, we used the intensity of NFM staining in the hilus, because the intensity in the subicular or other areas in the dentate gyrus was not stable.

## Dispersed dentate granule cell cultures

P3-*Kif2a*-cKO and WT mice were euthanized at P5, and their hippocampal dentate gyri were dissected (*Hagihara et al., 2009*). Each dentate gyrus was trypsinized and gently triturated to isolate cells ($3.5 \times 10^4$ cells/cm²), which were placed in a four-well glass chamber (Nunc, 155411). Chambers were coated with poly-L-lysine overnight at room temperature (Sigma, St. Louis, MO), washed with DW for 2 hr twice, and then coated with laminin overnight at 4°C (inquiry) to clearly visualize any morphological differences in the rapid growth of neurites. Dispersed cells were cultured in MEM (Gibco Thermo Fisher, MA)/Neuro Brew-21 (MACS, Bergisch Gladbach, Germany) at 37°C in a humidified atmosphere containing 5% $CO_2$. To confirm the characteristics of the cultured neurons, the cells were stained with anti-NFM, anti-MAP2, anti-Prox1, and anti-AnkyrinG antibodies.

## Organotypic hippocampal slice cultures

To confirm excitatory recurrent circuits in the tamoxifen-injected *Kif2a*-cKO hippocampus, organotypic hippocampal slice cultures were prepared as previously described (*Koyama et al., 2004*). Briefly, P4 mice were deeply anesthetized, and their brains were removed and cut into 300-μm-thick transverse slices using a VR-1200S (Leica Biosystems, Wetzlar, Germany) in a cold oxygenated Gey's balanced salts solution supplemented with 25 mM glucose. Entorhino-hippocampi were dissected and cultured using a membrane interface technique (*Stoppini et al., 1991*). Briefly, the slices were

placed on sterile 30-mm-diameter membranes (Millicell-CM; Millipore, Bedford, MA) and transferred to six-well tissue culture trays. The cultures were fed with 1 ml of 50% minimal essential medium (Invitrogen, Gaithersburg, MD), 25% horse serum (Cell Culture Lab, Cleveland, OH), and 25% HBSS and the cells were maintained in a humidified incubator at 37°C containing 5% $CO_2$. The medium was changed every 3.5 days.

## Acknowledgements

We thank Dr. Phil Soriano (Mt. Sinai School of Medicine) for characterizing the CBA-CreERt mice by using ROSA-STOP reporter mice, Mr. Ishidate for assisting with the confocal analysis, Ryuta Koyama (University of Tokyo) for technical suggestions, Prof. Yasuko Noda (Jikei Medical School) for the KIF2A monoclonal antibody and critical discussions, Mr. Yusuke Takahashi for helping statistical analysis, and all members of our laboratory for constructive discussions, technical support, and kind encouragement. The authors thank Ms. Synthia Ishizawa and Nature Research Editing Service for the English language review. This project was supported by a Grant-in-Aid for Specially Promoted Research and a Grant-in-Aid for scientific research from the Ministry of Education, Culture, Sports and Technology of Japan to N.H., a grant from The National Plan for Science, Technology and Innovation (MAARIFAH) – King Abdul-Aziz City for Science and Technology - the Kingdom of Saudi Arabia – award number (12-BIO3059-03), and a grant from the Deanship of Scientific Research (DSR: 1-6-1432/HiCi) from King Abdulaziz University Jeddah. The authors also, acknowledge with thanks Science and Technology Unit, King Abdul-Aziz University for technical support.

## Additional information

### Funding

| Funder | Grant reference number | Author |
|---|---|---|
| Ministry of Education, Culture, Sports, Science, and Technology | Grant-in-Aid for Scientific Research | Nobutaka Hirokawa |
| King Abdulaziz University | NSTIP Strategic Technologies Program Project (12-BIO3059-03) | Nobutaka Hirokawa |
| King Abdulaziz University | Deanship of Scientific Research (DSR: 1-6-1432/HiCi) | Muhammad Imran Naseer Adeel G Chaudhary Mohammed H Al-Qahtani |

The funders had no role in study design, data collection and interpretation, or the decision to submit the work for publication.

### Author contributions

Noriko Homma, Conceptualization, Resources, Data curation, Formal analysis, Visualization, Methodology, Writing—original draft; Ruyun Zhou, Data curation, Formal analysis; Muhammad Imran Naseer, Adeel G Chaudhary, Mohammed H Al-Qahtani, Resources; Nobutaka Hirokawa, Supervision, Funding acquisition, Project administration, Writing—review and editing

### Author ORCIDs

Noriko Homma    http://orcid.org/0000-0002-1322-9993
Nobutaka Hirokawa    http://orcid.org/0000-0002-0081-5264

### Ethics

Animal experimentation: This study was performed in strict accordance with the recommendations in the Guide for the Care and Use of Laboratory Animals of the Graduate School of Medicine, University of Tokyo. All of the animals were handled according to approved institutional animal care and use committee protocols of the University of Tokyo. The protocol was approved by the Committee on the Life Science Research Ethics and Safty of the Graduate School of Medicine, University of

Tokyo (Permit Number: Med-P10-130-133). All surgery was performed under sodium pentobarbital anesthesia, and every effort was made to minimize suffering.

### Decision letter and Author response

Decision letter https://doi.org/10.7554/eLife.30935.033
Author response https://doi.org/10.7554/eLife.30935.034

---

## Additional files

### Supplementary files

• Transparent reporting form
DOI: https://doi.org/10.7554/eLife.30935.031

---

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
