## [Decision Letter]

[Editors’ note: a previous version of this study was rejected after peer review, but the authors submitted for reconsideration. The first decision letter after peer review is shown below.]

Thank you for submitting your work entitled "KIF2A Regulates Establishment of Accurate Hippocampal Wiring and Its Loss Causes Severe Epilepsy" for consideration by *eLife*. Your article has been reviewed by two peer reviewers, and the evaluation has been overseen by Gary Westbrook as the Senior and Reviewing Editor. The reviewers have opted to remain anonymous.

Our decision has been reached after consultation between the reviewers. A summary of the reviewers' comments and the discussion are included below. We think this is a very interesting phenotype, bit there were aspects of the work that we think require additional data and that the work was somewhat overinterpreted. Thus we cannot consider the manuscript further at this time. However if you are able to address these issues with additional data and rewriting, we would be will consider a resubmission in the future. If you decide to resubmit please include a point-by-point response to the comments and be aware that there is no guarantee that a resubmitted manuscript would lead to acceptance.

Summary:

This is an interesting study examining the effects of conditional Kif2a deletion in the postnatal mouse brain. The authors accomplish this through the generation of a novel, floxed allele of KIF2A gene and crossing to a β-actin promoter tamoxifen-inducible Cre line. Doses of tamoxifen administered daily between P14 and P21 led to the loss of KIF2A expression in the brain and the onset of hyperactivity, weight loss and epilepsy. The dentate gyrus was particularly affected by KIF2A deletion where they observed misplaced granule cells and increased axon terminals in the outer molecular layer. In culture KIF2A-null cells exhibited minor neurites that were positive for both axon and dendrite markers and increased axon branching with the latter phenotype reversed by re-introduction of KIF2A. From these data, the authors conclude that KIF2A regulates hippocampal wiring, and loss of KIF2A results in aberrant sprouting of axon like neurites from granule cells that precipitates epileptiform activity in the hippocampus. The resulting phenotype is interesting in that there is a proliferation of terminals in ectopic locations within the dentate gyrus that is in some ways reminiscent of mossy fiber sprouting in TLE models and patients. Overall this is an excellent study, but key aspects are lacking or could use significant refinement to support the conclusions.

1) The authors claim in multiple places that there is a distinction between their model and animal TLE models since the sprouting occurs in their model without an initial inciting excitatory event. This is a distinction with uncertain accuracy. They show that their mice have spontaneous activity so it is hard to know how much is being driven by the ongoing abnormal excitation in these fibers. The treatment with VPA is perhaps a bit helpful for this, however, VPA likely acts via augmenting inhibitory synaptic activity and may not actually be affecting cell-intrinsic abnormal excitation in the DGC. Another even more important point is that the authors neglect the recent studies of Danzer and colleagues (and others) who have shown that mutants in PTEN develop spontaneous MF sprouting and seizures. I would say that the authors results are interesting and novel without making attempts to overstate the global novelty of a model that spontaneously develops sprouting.

2) The authors try to argue that "migration" is essentially complete by P15 and thus they are studying the formation of circuits instead. However, the dentate continues to generate new neurons and this is still ongoing at P15 at a very high level. These newborn cells are produced in the subgranular zone and migrate into the dentate. It seems quite possible that *Kif2a* is involved in their migration too. Thus, it is hard to conclude that the ectopic DGC are due to abnormal excitation or due to their abnormal axonal morphology. This could still be a primary migratory problem. The authors need to acknowledge this issue and/or provide data in support of their conclusions.

3) It is interesting that the authors note the phenotype of multiple axons, which is distinct from MF sprouting in epilepsy. In epilepsy the axons are usually sprouted from the main axon and there are still clear dendritic processes. This is distinct and interesting. However, this does perhaps make it less physiologically relevant to the human problem. This issue needs discussion to avoid overinterpretation.

4) Their section distinguishing between the terminals in pilo-TLE dentates isn't convincing. The example here is a fairly weakly sprouted TLE mouse compared to many animal TLE examples. Furthermore it is worth remembering that in TLE it isn't only the MF that are sprouted but many axons in the hippocampus. It is just easier to detect the MF sprouting because of Timm's stain. In their model it is possible that the MF sprouting occupies other areas simply because of other axon types are not sprouted in their model, leaving more "space" for MF sprouting. The authors need to be more cautious with these interpretations as the current formulation is not convincing.

5) The authors claim that their model has sprouting and epilepsy without an "inciting" epileptic event. This is a big chicken-egg problem. The authors aren't able to actually exclude that their mice are having small subclinical seizures from very early on. The use of VPA doesn't exclude this possibility. This is another example of overinterpretation.

6) The authors make an effort to confirm that cerebral cortex lamination is not affected by postnatal conditional deletion of KIF2A. Yet, no characterization is presented regarding the impact of KIF2A loss on granule cell neurogenesis and migration. This relates to KIF2A's established role in mitosis and migration.

7) The emergence of what the authors refer to as apical axons is a critical aspect of the report, yet it is not clear if the cells producing these processes are established or newly integrated granule cells. Another issue associated with the apical axons is the lack of a clear demonstration that these neurites are capable of forming presynaptic processes on other neurons (if only demonstrated in culture). The observations in Figure 3, while interesting, are not quantified thus making it difficult to conclude the significance of these finding either in magnitude or how often such events are observed.

8) Additionally, the manuscript needs significant compositional editing and proofing.

[Editors’ note: what now follows is the decision letter after the authors submitted for further consideration.]

Thank you for submitting your article "KIF2A regulates the development of dentate granule cells and postnatal hippocampal wiring" for consideration by *eLife*. Your article has been reviewed by two peer reviewers, and the evaluation has been overseen by Gary Westbrook as the Senior and Reviewing Editor. The following individual involved in review of your submission has agreed to reveal his identity: Samuel J Pleasure (Reviewer #1). The reviewers were very satisfied with this version of the manuscript. The study brings new insights both to the cell biology of KIF2A function in neurons during circuit assembly and to the cytoskeletal contributions to the establishment and maintenance of neural circuitry. As such, it will be a valuable contribution to multiple disciplines.

We ask only that you consider these minor suggestions before final acceptance:

1) Given that this is a novel mechanism for hyper-excitability, the manuscript would benefit from a discussion of this dendro-axonal conversion including whether this phenomenon may be a result of altered chemo-attraction/repulsion or could be cell autonomous.

2) The Abstract requires further refinement including where the olfactory bulb is referred to as a site of adult neurogenesis and the authors could better define 3w-*Kif2a*-cKO at this section of the manuscript.

3) The inducible Cre model and various control strains/treatments could also be more clearly described in the Results section.

4) Methodological choices remain unclear including the reasoning for culturing the hippocampal slices for 10 days rather than using acute slices for electrophysiological recordings and the switch from tau1 staining to NFM for axon labeling. Clarification would benefit the reader in following the logic of the study.

5) Immuno-histochemical examination of the distribution of the axon hillock marker Ankyrin G could be a straightforward and important way to better define the nature of the additional neurites observed in the 3w-*Kif2a*-cKO hippocampi as they are posited to contribute to the phenotype of the cKO.

6) The term "ictal mouse" could be replaced with "mouse during the ictal phase of seizure".

7) Figure 2 and Figure 3 legends refers to statistical measures no longer found in those figures.

---

## [Author Response]

[Editors’ note: the author responses to the first round of peer review follow.]

1) The authors claim in multiple places that there is a distinction between their model and animal TLE models since the sprouting occurs in their model without an initial inciting excitatory event. This is a distinction with uncertain accuracy. They show that their mice have spontaneous activity so it is hard to know how much is being driven by the ongoing abnormal excitation in these fibers. The treatment with VPA is perhaps a bit helpful for this, however, VPA likely acts via augmenting inhibitory synaptic activity and may not actually be affecting cell-intrinsic abnormal excitation in the DGC. Another even more important point is that the authors neglect the recent studies of Danzer and colleagues (and others) who have shown that mutants in PTEN develop spontaneous MF sprouting and seizures. I would say that the authors results are interesting and novel without making attempts to overstate the global novelty of a model that spontaneously develops sprouting.

In the original manuscript, we indeed overstated the differences between our model and animal TLE models. After learning about the research on PTEN, including the studies of Danzer and colleagues, we have changed a focus of our manuscript from the role of KIF2A in TLE to a focus on postnatal development of DGCs because spontaneous sprouting has not been determined to be a globally novel phenomenon.

We still describe the differences between our model and animal TLE models, but we have minimized the claims regarding these differences. As supporting evidence, we also used an anti-epileptic drug, but changed the drug from VPA to carbamazepine (CBZ), which is a blocker of voltage-gated sodium channels (Figure 4—figure supplement 1). The results indicated that axonal terminals were spread in the OML in CBZ-treated 3w-*Kif2A*-cKO, which was the same result as that observed in VPA-treated mice.

2) The authors try to argue that "migration" is essentially complete by P15 and thus they are studying the formation of circuits instead. However, the dentate continues to generate new neurons and this is still ongoing at P15 at a very high level. These newborn cells are produced in the subgranular zone and migrate into the dentate. It seems quite possible that Kif2a is involved in their migration too. Thus, it is hard to conclude that the ectopic DGC are due to abnormal excitation or due to their abnormal axonal morphology. This could still be a primary migratory problem. The authors need to acknowledge this issue and/or provide data in support of their conclusions.

As the reviewer stated, we should have tested DGC migration, because we have shown that KIF2A prenatally affects neuronal migration in the hippocampus, thus resulting in a laminary defect (Homma et al., 2003). Therefore, in this revised manuscript, we tested DGC migration by using birth dating analysis (Figure 3). We injected two thymidine analogues (CldU and IdU) into pups for 7 days before and during tamoxifen injection. In this experiment, CldU-positive cells were designed to lose KIF2A expression after cell proliferation, but IdU-positive cells were designed to lose KIF2A expression during cell proliferation. At P35, we fixed the brains, and brain slices were immunostained with specific antibodies for each dU. After measuring the number of dU-positive cells and the distance between the baseline of the GCL and the cell, we compared the values statistically. Significant differences were not detected in either cell proliferation (Figure 3) or cell migration (Figure 3). Therefore, we believe that cell proliferation and migration might contribute less to the development of an epileptic hippocampus than hippocampal wiring due to aberrant DGC development.

3) It is interesting that the authors note the phenotype of multiple axons, which is distinct from MF sprouting in epilepsy. In epilepsy the axons are usually sprouted from the main axon and there are still clear dendritic processes. This is distinct and interesting. However, this does perhaps make it less physiologically relevant to the human problem. This issue needs discussion to avoid overinterpretation.

In the revised Discussion, we have added some explanation of this issue (subsection “Future perspectives”). In human patients with mutant KIF2A, there has been no histological evidence of hippocampal phenotypes, because all of the patients are still living.

4) Their section distinguishing between the terminals in pilo-TLE dentates isn't convincing. The example here is a fairly weakly sprouted TLE mouse compared to many animal TLE examples.

We had a similar thought, and we repeated experiments more than three times to generate a pilocarpine-induced TLE animal model, but the results were the same, in that the axon terminals in IML were substantially weaker than those in many mouse TLE models. We surmise that three weeks (from postnatal week 3 to 5) is not a sufficient amount of time to develop severe recurrent circuits from sprouted axons.

Furthermore it is worth remembering that in TLE it isn't only the MF that are sprouted but many axons in the hippocampus. It is just easier to detect the MF sprouting because of Timm's stain. In their model it is possible that the MF sprouting occupies other areas simply because of other axon types are not sprouted in their model, leaving more "space" for MF sprouting. The authors need to be more cautious with these interpretations as the current formulation is not convincing.

Thank you for the constructive suggestion. We obtained additional data to compare the “space” in the molecular layer between the *Kif2a*-cKO and TLE animal models, and we now mention this possibility in the revised Discussion (subsection “The contribution of postnatal KIF2A loss to hippocampal wiring and epilepsy”, third paragraph). We believe that the “space” might not be different between 3w-*Kif2a*-cKO mice (Figure 4) and 3w-induced TLE animal models (Figure 4) because both mice showed TLE from the postnatal week 4 onward, and the sprouting of many axons, including MF in the hippocampus, might be induced by TLE in the molecular layer at the same level.

In our study, however, when we used anti-epileptic drugs such as carbamazepine (Figure 4—figure supplement 1), the Timm’s staining in the molecular layer was darker than that observed in untreated *Kif2A*-cKO. Therefore, as the reviewer mentioned, there might be more “space” in the molecular layer in CBZ-treated 3w-*Kif2A*-cKO than in untreated 3w-*Kif2A*-cKO.

5) The authors claim that their model has sprouting and epilepsy without an "inciting" epileptic event. This is a big chicken-egg problem. The authors aren't able to actually exclude that their mice are having small subclinical seizures from very early on. The use of VPA doesn't exclude this possibility. This is another example of overinterpretation.

We also have considered the “chicken-egg” problem in the previous manuscript. In the revised paper, we mainly focus on KIF2A function in DGC development, not the pathogenesis of TLE, and we have made fewer claims regarding our findings. As a phenotype of less epileptic episodes, we explain the results of anti-epileptic drug treatment. In the revised paper, we used CBZ instead of VPA because CBZ blocks the voltage-gated sodium channels of the DGC.

6) The authors make an effort to confirm that cerebral cortex lamination is not affected by postnatal conditional deletion of KIF2A. Yet, no characterization is presented regarding the impact of KIF2A loss on granule cell neurogenesis and migration. This relates to KIF2A's established role in mitosis and migration.

As the reviewer stated, we also wanted to determine the influence of KIF2A loss on granule cell neurogenesis and migration. We have added data from a birth-dating analysis to Figure 3. Within a period of 3 weeks after tamoxifen injection, however, we did not detect a significant difference in DGC neurogenesis and migration. In the future, we want to determine the influence of KIF2A loss on adult neurogenesis and migration by using a late-onset Cko model, such as 8w-*Kif2a*-cKO mice, because those mice can live for more than 6 months (unpublished data).

7) The emergence of what the authors refer to as apical axons is a critical aspect of the report, yet it is not clear if the cells producing these processes are established or newly integrated granule cells.

We are also eager to know whether the granule cells are “established or newly integrated”. We succeeded in labeling established cells born at E13.5 at the top of the GCL by using in utero electroporation, but we were not able to immunostain established neurites with either axonal markers or a dendritic marker. In addition, the signal was so weak at P35 that we could not observe spine morphology.

We succeeded in comparing DGC morphology at the top, middle, and bottom by using Thy1-GFP-expressing M-line transgenic mice (Figure 5). The top DGCs lost KIF2A expression after being established, the middle ones lost it during migration, and the bottom ones lost it during proliferation or migration. The results showed that the bottom cells developed aberrant protrusions or thin neurites (Figure 5), and the middle and the top ones developed recurrent axons (Figure 5). The dendrites of the top DGCs showed morphological changes, and the trunks became thinner and the spines became more unstable (Figure 5), even if the frequency of the spine was not significantly different.

Another issue associated with the apical axons is the lack of a clear demonstration that these neurites are capable of forming presynaptic processes on other neurons (if only demonstrated in culture). The observations in Figure 3, while interesting, are not quantified thus making it difficult to conclude the significance of these finding either in magnitude or how often such events are observed.

In the revised paper, the frozen slices were stained with synaptoporin, which is a marker of both presynapses and DGC axons. The average intensity in the molecular layer was higher in *Kif2A*-cKO mice than in control mice. We attempted for more than a year to detect triple-stained synapses with Thy1-GFP, presynaptic marker, and DGC marker; however, this approach was unsuccessful in the slice samples, owing to technical issues (especially Thy1-GFP-expressing cells, which could not be stained with any antibodies). Ultimately, we did not add the data to this revised paper.

8) Additionally, the manuscript needs significant compositional editing and proofing.

We apologize for having submitted our manuscript with insufficient editing and proofreading. We have obtained English proofreading through the “Nature Research Editing Service”, which is one of the most-trustworthy editing groups worldwide. We believe the English in the recent version of the manuscript has been improved.

[Editors' note: the author responses to the re-review follow.]

We ask only that you consider these minor suggestions before final acceptance:1) Given that this is a novel mechanism for hyper-excitability, the manuscript would benefit from a discussion of this dendro-axonal conversion including whether this phenomenon may be a result of altered chemo-attraction/repulsion or could be cell autonomous.

In the revised Results section, we have added some comments on this issue (subsection “Cultured *Kif2a*-cKO DGCs showed dendro-axonal conversion from DIV3”, end of first paragraph), starting that the dendro-axonal conversion is likely a result of a cell autonomous process rather than altered chemo-attraction/repulsion because dispersed cultured DGCs showed this phenotype without chemical treatment (Figure 6).

2) The Abstract requires further refinement including where the olfactory bulb is referred to as a site of adult neurogenesis and the authors could better define 3w-Kif2a-cKO at this section of the manuscript.

Thank you for these helpful suggestions. In the revised Abstract, we have considered your comment and finally decided to delete the reference to the “olfactory bulb”, as it is not the focus of this manuscript. We also deleted “3w” from “*3w-Kif2a-cKO”*, because we not only used *3w-Kif2a-cKO* mice, but also *P3-Kif2a-cKO* mice for cultured hippocampal slice (Figure 2) and cultured dentate granular cells (Figure 6, Figure 6—figure supplement 1, and Figure 6—figure supplement 2).

3) The inducible Cre model and various control strains/treatments could also be more clearly described in the Results section.

We apologize for submitting our manuscript with insufficient information on the inducible Cre model and various control strains/treatments. In the revised Results section, we have added clear explanation of the models and treatments used here (subsection “Weight loss, hyperactivity, and severe epilepsy were exhibited by 3w-*Kif2a*-cKO mice”, first paragraph..

4) Methodological choices remain unclear including the reasoning for culturing the hippocampal slices for 10 days rather than using acute slices for electrophysiological recordings and the switch from tau1 staining to NFM for axon labeling. Clarification would benefit the reader in following the logic of the study.

We apologize for submitting our manuscript with insufficient information. In the revised Results section, we have added a clear explanation of the methods used (subsection “The epileptic hippocampus was developed in 3w-*Kif2a*-cKO mice”, last paragraph).

5) Immuno-histochemical examination of the distribution of the axon hillock marker Ankyrin G could be a straightforward and important way to better define the nature of the additional neurites observed in the 3w-Kif2a-cKO hippocampi as they are posited to contribute to the phenotype of the cKO.

Thank you for this constructive suggestion. We obtained additional data to compare the distribution of ankyrin G between the WT and P3-*Kif2a*-cKO mice at DIV3 (Figure 6—figure supplement 1). As the reviewer and we expected, there were more ankyrin G-positive neurites in dispersed DGCs (Figure 6—figure supplement 1).

6) The term "ictal mouse" could be replaced with "mouse during the ictal phase of seizure".

As per the reviewer’s suggestion, we have replaced the term "ictal mouse" with "mouse during the ictal phase of seizure".

7) Figure 2 and Figure 3 legends refers to statistical measures no longer found in those figures.

Thank you for this helpful comment. We have added a graph to each figure (Figure 2 and Figure 3).